# Rapid and reversible optogenetic silencing of synaptic transmission by clustering of synaptic vesicles

Dennis Vettkötter[1,2], Martin Schneider [1,2,3], Brady D. Goulden[4], Holger Dill[1,2], Jana Liewald [1,2], Sandra Zeiler[1,2], Julia Guldan[1,5], Yilmaz Arda Ateş[1,5], Shigeki Watanabe [4] & Alexander Gottschalk [1,2] ✉

Acutely silencing specific neurons informs about their functional roles in circuits and behavior. Existing optogenetic silencers include ion pumps, channels, metabotropic receptors, and tools that damage the neurotransmitter release machinery. While the former hyperpolarize the cell, alter ionic gradients or cellular biochemistry, the latter allow only slow recovery, requiring de novo synthesis. Thus, tools combining fast activation and reversibility are needed. Here, we use light-evoked homo-oligomerization of cryptochrome CRY2 to silence synaptic transmission, by clustering synaptic vesicles (SVs). We benchmark this tool, optoSynC, in *Caenorhabditis elegans*, zebrafish, and murine hippocampal neurons. optoSynC clusters SVs, observable by electron microscopy. Locomotion silencing occurs with $tau_{on}$ ~7.2 s and recovers with $tau_{off}$ ~6.5 min after light-off. optoSynC can inhibit exocytosis for several hours, at very low light intensities, does not affect ion currents, biochemistry or synaptic proteins, and may further allow manipulating different SV pools and the transfer of SVs between them.

Neurons are specialized cells that transmit intercellular information via electrical and chemical signals. In terminals of chemical synapses, three kinds of synaptic vesicle (SV) pools are distinguished: reserve pool (RP), readily releasable pool (RRP), and recycling pool[1,2]. New SVs, filled with transmitter, reside in the RP. During the SV cycle, SVs can be recruited from the RP to the active zone plasma membrane (PM). Following arrival of an action potential (AP), $Ca^{2+}$ enters the terminal via voltage-gated $Ca^{2+}$ channels (VGCCs), which triggers SV fusion with the PM, facilitated by SNARE proteins, and leading to release of neurotransmitters into the synaptic cleft[3]. Current models describe the processes underlying chemical synaptic transmission by the steps of SV docking, priming, and fusion/exocytosis[4]. Some SV proteins remain clustered in the PM, thus facilitating their recycling by ultrafast endocytosis. The formation of new SVs from synaptic endosomes and

their refilling with neurotransmitters within the RP concludes the SV cycle[5,6].

To study nervous system function and underlying molecular and cellular processes, the ability to modulate neuronal activity is instrumental. Ideally, methods allowing such modulation reversibly, are applicable in intact animals. A major development in this context are techniques to influence neuronal activity with light, subsumed under the term optogenetics[7,8]. Light-responsive proteins are expressed in cells to affect their physiology in various ways. The first optogenetic protein used was *Chlamydomonas* channelrhodopsin-2 (ChR2), a blue light-gated cation channel[9]. When expressed heterologously, blue light application, within a few ms, induced depolarization of cultured mammalian neurons to trigger AP-driven synaptic transmission, and in intact *Caenorhabditis elegans* (*C. elegans*), even behavior could be

[1]Buchmann Institute for Molecular Life Sciences, Goethe University, D-60438 Frankfurt, Germany. [2]Institute of Biophysical Chemistry, Goethe University, D-60438 Frankfurt, Germany. [3]Max Planck Institute for Neurobiology, D-82152 Martinsried, Germany. [4]Department of Cell Biology and Solomon H. Snyder Department of Neuroscience, Johns Hopkins University, Baltimore, MD 21205, USA. [5]Master Program Interdisciplinary Neurosciences, Department of Biological Sciences, Goethe University, Frankfurt, Germany. ✉e-mail: a.gottschalk@em.uni-frankfurt.de

induced[10,11]. Many labs have adapted ChR2 to interrogate circuit-behavior relationships in diverse model organisms. By turning off illumination, the neuronal excitation is rapidly terminated (typically within 20–100 ms), thus ChR2-based optogenetics is highly reversible.

Besides excitation, inhibition informs about the function of neurons within circuits. Several silencing strategies using genetically encoded tools with different biophysical properties were developed over the past ca. 15 years[12]. Light-activated ion-pumps and -channels enable reversible hyperpolarization of neurons in milliseconds, thus resulting in suppression of neuronal activity with high spatial and temporal resolution[13–16]. However, many of these tools exhibit a decline in efficacy when stimulated over a prolonged period[12,17,18]. The inhibitory action of anion channelrhodopsins (ACRs) depends on membrane potential and the chloride gradient, which may be exhausted during prolonged (seconds to minutes) applications[19]. In synaptic terminals, due to their specific Cl⁻ gradient, ACRs can cause depolarization instead of hyperpolarization[19,20]. More recently, light-triggered G-protein-coupled receptors (GPCRs), activating $G\alpha_{i/o}$ pathways, were established for silencing[21–24]. Compared to ion channels, GPCR signaling to downstream inhibitory components is of slower onset kinetics (dozens of ms to sec)[12,21–24]. When GPCRs are to be used in a new cell type, G-protein coupling specificity may require (re)confirmation.

As an alternative to ionotropic or metabotropic silencing, other tools were developed that damage or degrade proteins essential for synaptic release. miniSOG (miniature singlet oxygen generator) produces reactive oxygen species upon blue-light stimulation, and these radicals oxidize susceptible amino acids such as cysteine, histidine, methionine, tryptophan, and tyrosine[25–28]. When miniSOG is attached to SV proteins like synaptobrevin (VAMP2) or synaptophysin (SYP1), application of blue light for short periods (seconds to minutes) leads to the inactivation of the SNARE complex or other components of the SV fusion machinery[27]. This is reversible, however, only at hours' time scale, as it requires de novo protein synthesis. Also, generation of damaging radicals has off-target effects on other (proximal) synaptic proteins, and also on proteins in the secretory pathway. Therefore, longer-lasting effects on synaptic strength and neuronal cell biology are conceivable but not well understood. To avoid off-target effects, a photosensitive degron (PSD) was adapted for applications in the nervous system. The PSD enables the degradation of specific proteins, triggered by light[29]. Fused to synaptic proteins, it allows higher precision for targeting of the damaging effects on single synaptic protein species[28]. Light-induced degradation of synaptotagmin (SNT-1) resulted in inhibition of neurotransmission to an extent comparable to miniSOG. However, this approach works only in the absence of endogenous SNT-1 (i.e. null mutants), or the PSD must be inserted in the genomic locus. Another approach targeting the SNARE complex is photoactivatable botulinum neurotoxin (PA-BoNT)[30], which cleaves VAMP2 in a light-dependent manner. PA-BoNT does not require constant illumination for long-lasting effects. Yet, as for miniSOG or PSD, reversibility through de novo protein synthesis takes up to 24 h. While miniSOG requires some minutes for full effect, PSD and PA-BoNT are effective after 30–60 min stimulation, and thus, their onset is rather slow.

Hence, there is still a demand for silencing tools that do not damage cellular proteins or alter cellular biochemistry, work with comparably high spatial and temporal precision, with onset and recovery kinetics relevant for synaptic transmission and behavior, and with sustained silencing qualities. Since the SV cycle and chemical synaptic transmission require mobilization and moving of SVs toward the active zone (AZ) membrane, sequestering of SVs may inhibit synaptic transmission. This might be achieved via light-induced protein clustering. *Arabidopsis thaliana* Cryptochrome-2 (CRY2), which can undergo light-dependent di- or oligomerization[31], is the most widely used cryptochrome in optobiological studies. Stimulation of CRY2 with blue light (~450 nm) induces homo-oligomerization[31,32], or

hetero-dimerization with the cryptochrome-interacting basic helix-loop-helix protein 1 (CIB1)[33,34]. These interactions were mapped to the photolyase homology region (PHR), containing the chromophore FAD. Thus, like other dimerization tools, such as iLID[35] or Magnets[36], CRY2 can act as a light-inducible dimer. However, it can also act as a single-component system. Previous studies utilized CRY2 heteromerization to trap target proteins into complexes using light[37]. This 'light-activated reversible inhibition by assembled trap' (LARIAT) was recently used to interfere with synaptic transmission by targeting VAMP2[38]. The authors suggested that their approach inhibits synaptic transmission by blocking the SV release machinery, i.e. somewhat similar to the PSD and miniSOG approaches. However, the homo-oligomerizing properties of CRY2 may enable cross-linking of SVs already before they reach the PM, thus inhibiting synaptic transmission. Furthermore, such an approach may allow investigation of various aspects of the SV cycle, if SV mobilization from the RP or their transport to the AZ is inhibited.

Here, we used a variant of CRY2, CRY2olig(535), combined with the homo-oligomerization inducing mutation E490G, as well as a truncation that reduces dark activity, and fused it to the SV protein synaptogyrin (SNG-1), a tetraspan vesicle membrane protein[39]. This yielded optoSynC, a tool for **opto**genetic **syn**aptic vesicle **c**lustering. By behavioral and electrophysiological assays, we show that optoSynC can efficiently inhibit synaptic transmission in different subtypes of neurons of *C. elegans* within seconds, allows for long-term silencing for several hours, and can recover in the absence of light within minutes. Furthermore, optoSynC allowed abolishing escape behavior in zebrafish, and effectively eliminated synaptic transmission in murine hippocampal neurons. By electron microscopy, we demonstrate that SVs show marked clustering in response to light activation. Thus, optoSynC blocks synaptic transmission, at least in part, by impeding SV mobility.

## Results

### Development of optoSynC, an optogenetic tool for synaptic vesicle clustering

To interfere with synaptic transmission by SV clustering, we first tried a LARIAT-based approach, utilizing the heterodimerization of CRY2 with its interaction partner CIB1. Synaptogyrin is an abundant SV protein[40], whose loss-of-function neither results in major defects in synaptogenesis nor neuronal activity in *C. elegans*[39]. Functions in clathrin-independent endocytosis were observed, but only in combination with other mutations[41]. Therefore, SNG-1 can serve as an 'inert' anchor to attach proteins to the SV, as we did previously for PA-BoNT[30]. Enhanced CRY2 variants with C-terminal charge alteration, e.g. E490G, promote light-induced homo-oligomerization; this tool was termed CRY2olig[42]. Truncation, as in CRY2(535), improves expression and reduced self-association in the dark[43]. We fused CIBN, the N-terminal portion of CIB1, tagged with GFP, to SNG-1 panneuronally, and expressed an mOrange2-tagged CRY2olig(535) in the cytosol. This design resembles the recently reported opto-vTrap[38]. Blue light illumination (470 nm) should activate the PHR domain of CRY2 (refs. 44–46), inducing dimerization, and may thus trap SVs in clusters and inhibit neurotransmission (Supplementary Fig. 1a). We tested this by swimming behavior, which is sensitive to malfunction of the motor neurons. However, even prolonged illumination (15 min) did not alter swimming (Supplementary Fig. 1b, c). Thus, the LARIAT approach appears not to work in *C. elegans*.

We therefore turned to homo-oligomerization of CRY2[47]. Though this mechanism is complex and only partially characterized[48], many CRY2 optogenetic tools rely on it[32,42,49]. We fused CRY2olig(535) to the C-terminus of SNG-1 and introduced the construct, hereafter termed 'optoSynC' (optogenetic synaptic vesicle clustering), in *sng-1(ok234)* null mutants, expressing it from a pan-neuronal promoter. Light application-induced oligomerization may trap SVs in clusters (Fig. 1a), or block release sites in the PM, thus perturbing neuronal activity one

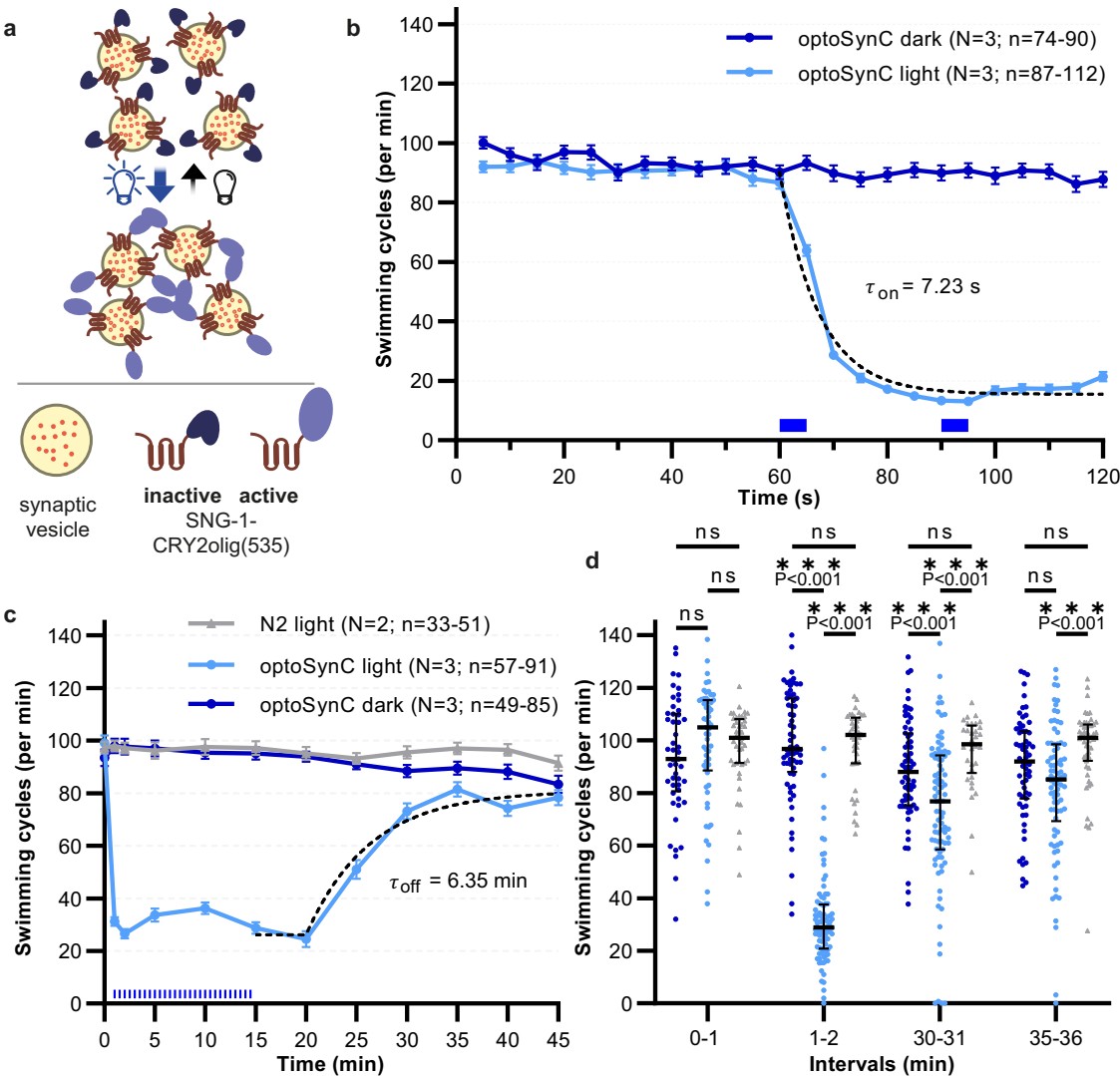

**Fig. 1 | OptoSynC inhibits behavior within seconds and recovers within minutes in the dark. a** Schematic illustrating SV clustering through homo-oligomerization of CRY2olig(535) upon blue light illumination. **b** Mean (±s.e.m.) swimming cycles of worms expressing optoSynC pan-neuronally. Illumination (470 nm, 0.1 mW/mm², 5 s / 25 s ISI) is indicated by blue rectangles; dotted line: one-phase decay fit (60 –120 s). **c** As in **b**, longer time course. Sustained inhibition of swimming by ongoing light pulses, and recovery in the dark. Dotted line: 'plateau followed by one phase association'-fit. N2 – non-transgenic wild type. **d** Group data (speed of individual animals, median + inter-quartile range) before (0–1 min), during (1–2 min), and after (30–31; 35–36 min) blue illumination. Two-way ANOVA with Bonferroni correction between light and dark measurements of wild type and optoSynC expressing animals (*sng-1(ok234)* background); ***$p < 0.001$; ns – non-significant. Number of individual animals (*n*) from left to right: 44, 57, 49, 42, 91, 64, 33, 85, 65, 51, 82, 63; across $N = 2$ (wild type) and $N = 3$ (optoSynC) independent experiments with animals picked from independent populations; range of individual animals across each measured time point and over $N$ independent experiments is indicated for **b** and **c**.

way or another. To explore this, we recorded swimming cycles under dark and light conditions (Fig. 1b, c). Activation of optoSynC (470 nm, 0.1 mW/mm², 5 s) nearly abolished swimming behavior (a decrease by 80%) within the first 20 s after one light pulse ($\tau_{on}$ = 7.23 s; Fig. 1b, d and Supplementary Movie 1). Application of further light pulses (5 s/25 s inter-stimulus interval - ISI) maintained the inhibition of swimming cycles for minutes. After illumination, animals recovered normal swimming behavior within 15–20 min (Fig. 1c, d; $\tau_{off}$ = 6.35 min). This indicates that the optoSynC construct returned to the dark state, likely releasing SVs from putative clusters. optoSynC can be used repeatedly for inactivation and recovery of neuronal activity (Supplementary Fig. 2a).

We characterized optoSynC efficacy under different illumination protocols and light intensities. Using continuous vs. pulsed light slightly increased the speed of the effect on swimming behavior (Supplementary Fig. 2b). When we reduced the light intensity from 0.1 mW/mm² to 1.4 µW/mm², the effects developed more slowly, and continuous illumination was more efficient than a single light pulse. A

light titration curve showed that 4.3–6.5 µW/mm² were sufficient to achieve maximal effects with 4 pulses of light (Supplementary Fig. 2c–e). To facilitate visualization of the SV cloud via optoSynC, we inserted a fluorescent protein between SNG-1 and CRY2olig(535). However, this reduced functionality by 50% (Supplementary Fig. 3a–c). In sum, optoSynC is a highly sensitive tool that inhibits locomotion within 10–20 s after light-stimulation, likely by affecting synaptic transmission, and recovers within 15–20 min in the dark.

**OptoSynC photoactivation can alter behavior within seconds**

OptoSynC activation inhibits locomotion, likely by blocking SV mobility and/or fusion. Since analysis of swimming behavior requires at least 10–20 s of data, it is not suited to precisely determine how fast optoSynC may act. We thus also analyzed crawling locomotion speed (Fig. 2). The short-wavelength photoreceptor LITE-1 mediates an escape response[50], which results in an increased crawling speed upon illumination. We analyzed animals expressing optoSynC; non-transgenic wild type and *sng-1(ok234)* were used as controls,

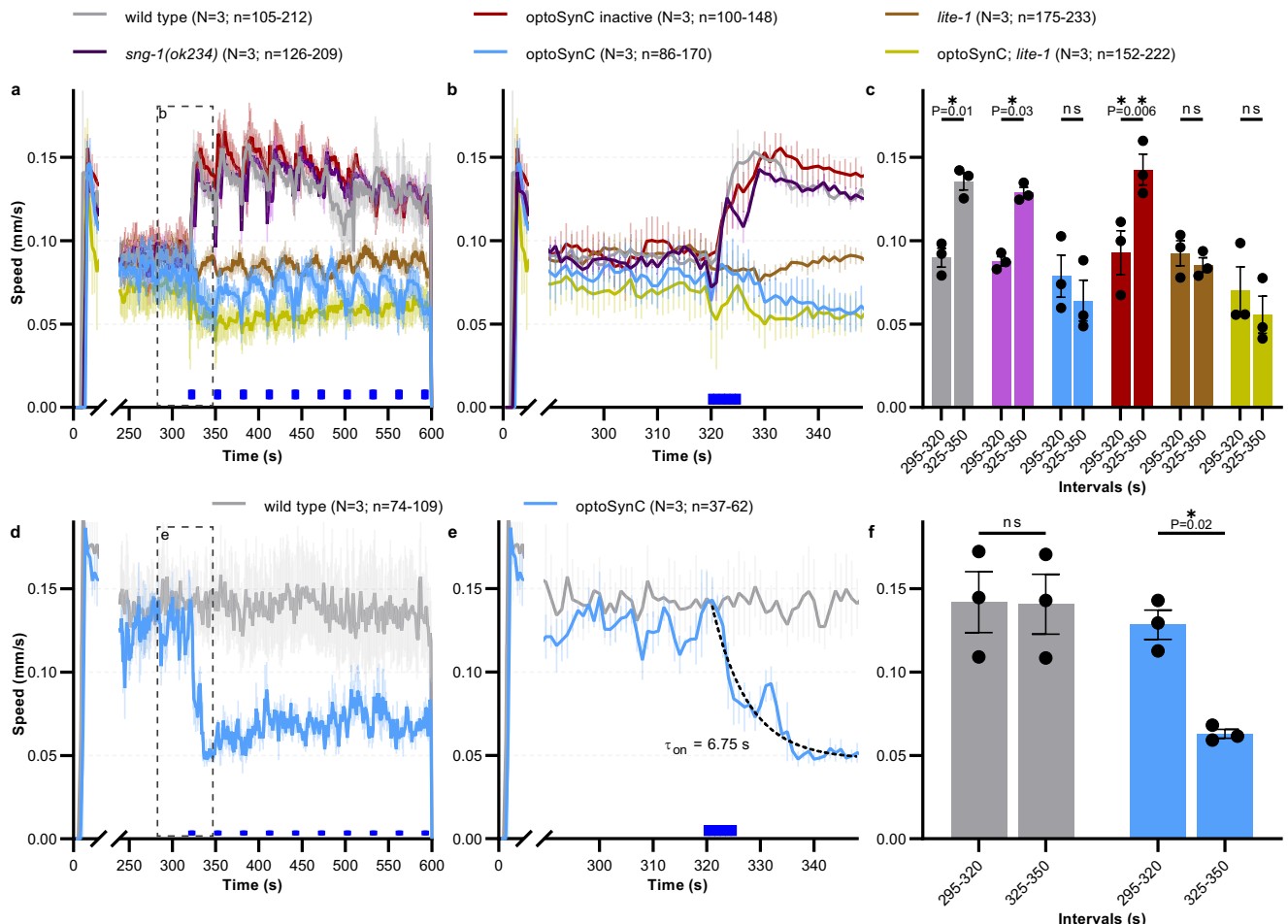

**Fig. 2 | Immediate inhibition of the LITE-1-dependent escape response by optoSynC. a** Mean ± s.e.m crawling speed analysis, blue light application (470 nm, 1 mW/mm², 5 s / 25 s ISI, indicated by blue rectangles), genotypes as indicated (optoSynC expressing animals are in *sng-1(ok234)* background). **b** Close-up of box indicated in **a**. **c** Group data, mean crawling speed of animals tested in *N* = 3 independent experiments (±s.e.m), analyzed as mean of time intervals before (295–320 s), and after (325–350 s) first light pulse; number of independent animals (*n*) across all independent experiments (*N*, i.e. animals picked from *N* independent populations) is indicated as range. **d** As in **a**, but using only 0.1 mW/mm² stimuli. **e** Close-up of box indicated in **d**. Dotted line represents one phase decay fit. **f** As in **c**, for data in **d**. In **c**, **f**, data are statistically analyzed with two-way ANOVA, Bonferroni correction, ns not significant.

transgenic animals are in *sng-1(ok234)* background, unless otherwise stated. To evoke a robust escape response, we illuminated animals with blue light pulses (5 s / 25 s ISI, 1 mW/mm²; much more than required to activate optoSynC, Fig. 1 and Supplementary Fig. 2). Crawling speed of wt and *sng-1(ok234)* animals rapidly increased by ~60% within 5 s, and decayed again after the light was turned off. However, animals expressing optoSynC (also activated by blue light), were unable to accelerate (Fig. 2a–c and Supplementary Movie 2). Already the response to the first light pulse was largely attenuated, and speed did not increase, also for subsequent stimuli, while non-transgenic animals always markedly increased their speed. *lite-1(ce314)* animals, which are unable to detect blue light, did not show escape behavior. Thus, activated optoSynC likely inhibits the escape response instantly.

When the escape response was avoided at lower light intensity (0.1 mW/mm²) optoSynC activation significantly decreased the basal crawling speed by ca. 60% ($\tau_{on}$ = 6.75 s; Fig. 2d, e). To confirm that optoSynC expression has no adverse effects on behavior, we mutated the FAD-binding pocket of CRY2 at position D387A, rendering CRY2 photoinactive[34]. Compared to *sng-1(ok234)* animals, animals expressing the resulting optoSynC-DA responded similarly to the blue light pulses (Fig. 2a–c). Thus, expression of optoSynC does not per se affect synaptic transmission, unless it is photo-activated.

## optoSynC activation strongly decreases mPSC frequency and can reduce cholinergic transmission for hours

To more directly examine the effect of activated optoSynC on SV release, we recorded miniature post-synaptic currents (mPSCs) from muscle cells, which are innervated by motor neurons expressing optoSynC. mPSC frequency was significantly reduced in response to optoSynC activation (0–30 s: 33.29 ± 4.74 s⁻¹; 95–105 s: 13.35 ± 2.27 s⁻¹), as compared to wt (0–30 s: 33.05 ± 6.09 s⁻¹; 95–105 s: 34.84 ± 6.77 s⁻¹; Fig. 3a–d). This indicates a rapid and robust decrease of SV fusion events upon activation of optoSynC, possibly by clustering SVs. mPSC amplitude remained unchanged upon illumination (Fig. 3e, f).

Since mPSC amplitude is determined by loading of SVs with neurotransmitters and/or by SV size[51], these results suggest that optoSynC has no effect on SV properties. Patch-clamp recording of *C. elegans* muscles requires dissection of the animal, and the preparation can be recorded reliably only for a few minutes[52]. Therefore the approach is not suited for analysis of long-term effects of optoSynC on cholinergic transmission and its potential for long-term inhibition. Thus, we used a pharmacological assay in intact animals. Incubating animals in the acetylcholinesterase inhibitor aldicarb causes accumulation of acetylcholine (ACh) in the synaptic cleft, thus leading to progressive paralysis[53]. We compared animals with and without pan-neuronal optoSynC, constantly stimulated with low-intensity light

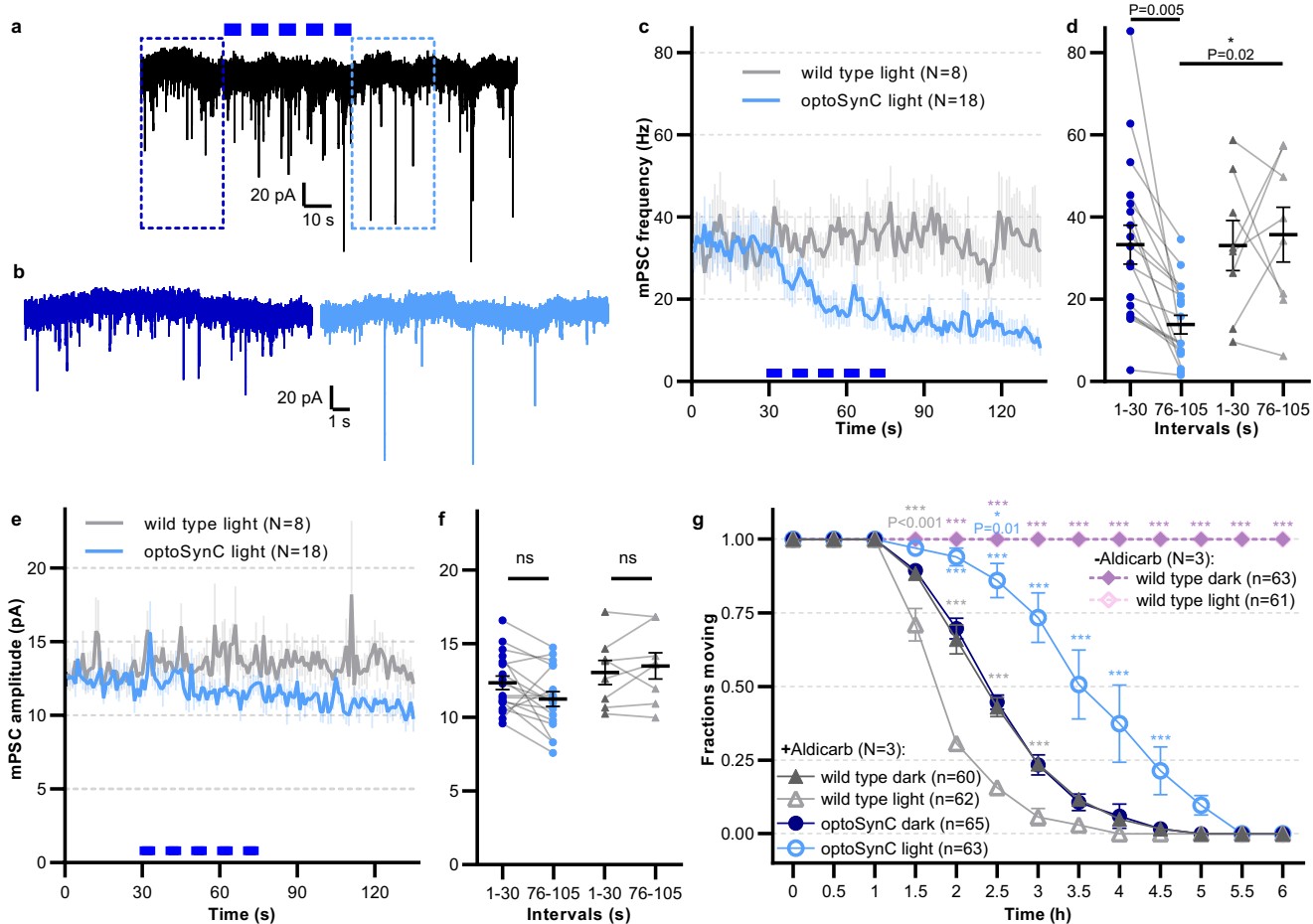

**Fig. 3 | optoSynC activation reduces miniature post-synaptic current (mPSC) rate at the neuromuscular junction (NMJ) and can block cholinergic transmission for hours. a** Representative postsynaptic current traces recorded in body wall muscle cells of optoSynC-expressing animals; blue light pulses (470 nm, 8 mW/mm², 5 s / 5 s ISI) indicated by blue rectangles. **b** Close-up of regions indicated in **a**, before (0–10 s, dark blue) and after (95–105 s, light blue) blue light illumination. **c** Mean (±s.e.m.) mPSC frequency with blue light illumination of wild type and optoSynC expressing animals. **d** Group analysis of data (mean ± s.e.m.) in

**c**, intervals before (0–30 s) and after (76–105 s) illumination. Number of independent animals is $n = 8$ (wild type) and $n = 18$ (optoSynC). **e, f** Analysis of mPSC amplitude (mean ± s.e.m.), as in **c, d. g** Paralysis of animals in response to 2 (or 0) mM aldicarb under continuous blue light illumination (470 nm, 0.05 mW/mm²), or in the dark, as indicated (mean ± s.e.m.). Two-way ANOVA with Bonferroni correction of the indicated number of animals (in **b**–**f**), or of $N = 3$ experiments averaged at the indicated time points with $n$ independent animals as indicated (in **g**). ***$p < 0.001$, ns not significant.

(470 nm, 0.05 mW/mm²) or kept in the dark (Fig. 3g). Animals kept in darkness throughout the experiment were paralyzed at the same rate as non-transgenic controls, while animals kept without aldicarb did not paralyze, neither in the dark, nor when illuminated for up to 6 h. Blue light stimulated animals expressing optoSynC paralyzed significantly more slowly than controls, most likely due to inhibition of ACh release. In contrast, wild type animals stimulated with blue light paralyzed significantly faster than dark controls, likely due to the LITE-1 mediated escape response (Fig. 2a), increasing ACh release.

### Ultrastructural analysis in optoSynC synapses unravels light-induced SV clustering

The behavioral and physiological effects of optoSynC activation could result from clustering of SVs, which may thus not be mobilized from the RP. Alternatively, proteins of the SV, particularly oligomerized SNG-1::CRY2olig(535), may remain in the AZ membrane, instead of being recycled by ultrafast endocytosis[54–57]. This could prevent SV recycling, or docking and priming of further SVs during ongoing stimulation. To examine the mode of action of optoSynC in detail, we analyzed the ultrastructure of cholinergic synapses expressing optoSynC by serial section transmission electron microscopy (TEM)[51,54,56]. Animals were illuminated for 5 s (+light; 470 nm, 0.1 mW/

mm²) and high-pressure frozen (HPF) 25 s later, to ensure maximal inhibition of synaptic transmission (Fig. 1b and Supplementary Fig. 2d). Control animals (-light) were always kept in darkness. HPF samples were freeze-substituted, stained, and 40 nm thin sections analyzed. Plasma membrane (PM), dense projection (DP), cytosolic SVs, docked/tethered SVs, dense core vesicles (DCVs), and large vesicles (LVs) were annotated using SynapsEM software[58] (Fig. 4a, b and Supplementary Fig. 4a, b).

To explore whether SVs became clustered, we quantified the distance of each SV to its nearest neighboring SV in the same micrograph (Fig. 4c and Supplementary Fig. 4c, d). Photostimulated synapses showed significantly smaller nearest distances (here and below, 75–25 percentiles (inter-quartile range, IQR) are given: 42.30-20.28 nm, 26.55 nm median) when compared to unstimulated controls (50.82-23.35 nm, 31.01 nm median; Fig. 4c, d). To exclude that these effects were overestimated by analyzing individual SVs, we also calculated the mean of the nearest distances per micrograph (+light: 45.6–31.45 nm, 37.20 nm median; -light: 54.94-40.32 nm, 47.10 nm median; Fig. 4e), and per synapse (i.e. in consecutive sections; +light: 40.65-33.98 nm, 39.24 nm median; -light: 51.85-45.42 nm, 47.81 nm median; Fig. 4f). Both analyses confirmed that there is a significant decrease of distance between SVs in the photostimulated samples, arguing that SVs became

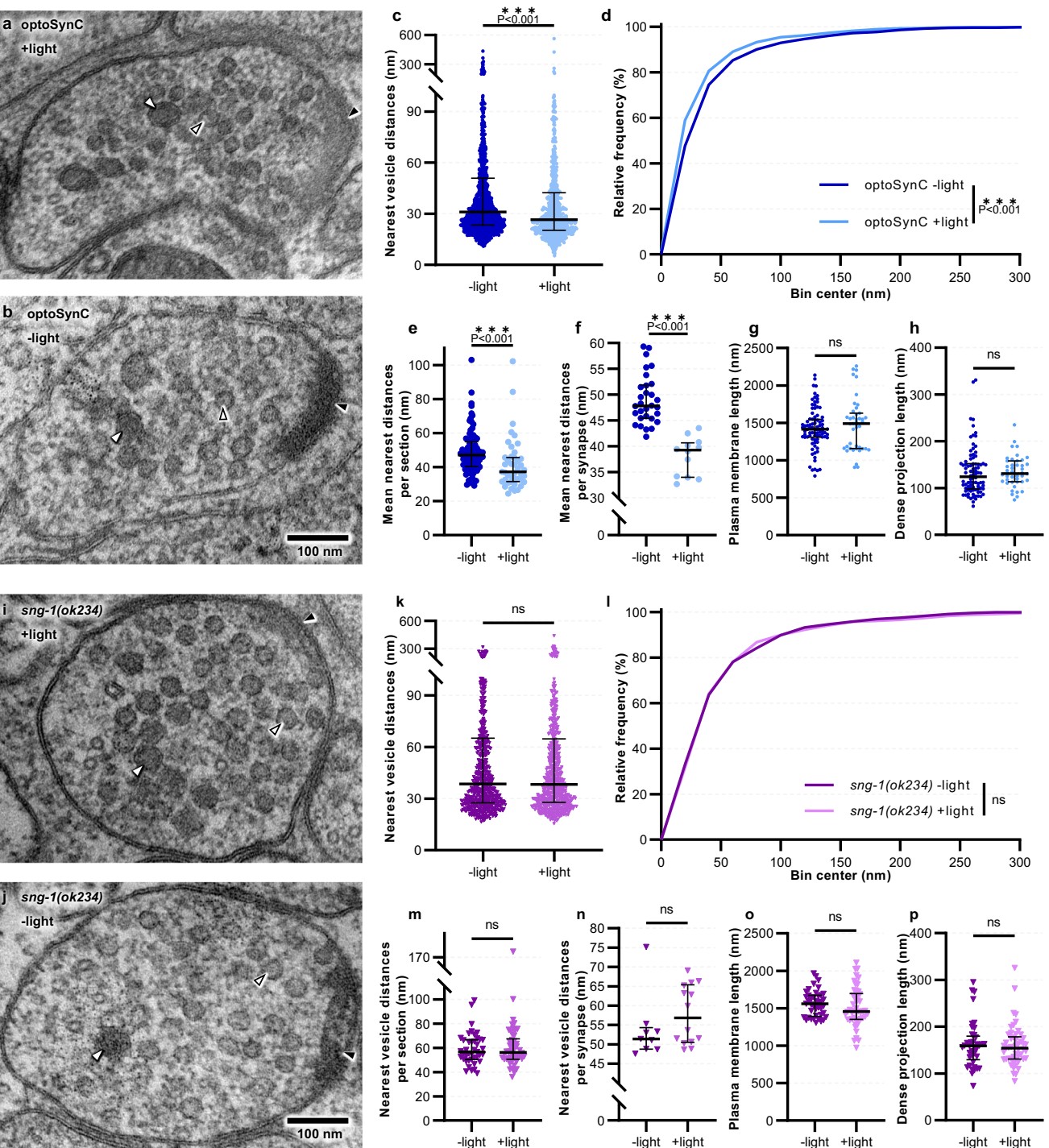

**Fig. 4 | optoSynC activation causes clustering of SVs at the ultrastructural level. a, b** Transmission electron micrographs of representative cholinergic synapses from animals illuminated for 5 s (**a**, +light) with blue light (470 nm, 0.1 mW/mm²), or kept in darkness (**b**, −light) before high-pressure freezing. SVs (open black arrowheads), dense core vesicles (DCVs, white closed arrowheads), and dense projection (DP, closed black arrowheads) are indicated. **c** Distance analysis of nearest vesicles for each analyzed cholinergic micrograph; −light ($n = 1473$), +light ($n = 819$). **d** Relative frequency distribution of nearest vesicle distances shown in **c**. **e** Mean nearest distances per section; −light ($n = 88$), +light ($n = 43$). **f** Mean nearest

distances per synapse; −light ($n = 30$), +light ($n = 12$). **g, h** Lengths of the PM and DP, respectively; −light ($n = 88$), +light ($n = 43$). Data in **c, e–h** are shown as median with 75–25% interquartile range (IQR). **i–p**, As for **a–h**, respectively, but for non-transgenic *sng-1(ok234)* mutant animals ($n = 707, 41, 9$ for −light, and $733, 40, 14$ for +light; in **k, n, l, o, m, p**, respectively). Sections originated from two animals, and 9–30 synapses for each condition. Statistical test used: Mann–Whitney (two-tailed) in **c, e, g–h, k, m–p**; unpaired *t*-test (two-tailed) in **f**; Kolmogorov–Smirnov in **d, l**; ***$p < 0.001$, ns not significant.

markedly more clustered by optoSynC. This occurs in addition to SV clustering in the RP, mediated by synapsin and the actin cytoskeleton[59]. So far, we pooled the data for all SVs, i.e. those in the RP (cytosolic) and those SVs tethered and docked to the PM. When we

restricted our analysis to the RP, we observed the same significant decrease of nearest SV distances following optoSynC activation (Supplementary Fig. 4c–f), while distances of docked SVs were not affected (Supplementary Fig. 4g–j). To control for non-specific effects of light,

i.e. in the absence of optoSynC, we also analyzed SV distances in non-transgenic *sng-1(ok234)* synapses, without and with light (Fig. 4i–n and Supplementary Fig. 4k–r). No significant differences were observed, thus, we conclude that optoSynC clusters SVs specifically.

Might stimulated optoSynC affect synaptic properties also due to protein aggregation in the SV or the PM? To this end we assessed features such as PM circumference (as a proxy for synapse size), length of the DP, SV diameter, or total SV number per section. No differences of PM perimeter or DP length were observed in photo-stimulated synapses (Fig. 4g, h). However, the diameters of SVs (+light: 36.46-20.61 nm, 38.25 nm median; −light: 33.40-29.25 nm, 31.18 nm median) and DCVs (+light: 52.37-44.45 nm, 49.43 nm median; -light: 49.62-42.67 nm, 46.37 nm median) were significantly larger after stimulation for optoSynC animals, but not for *sng-1(ok234)* controls (Supplementary Fig. 5a, b, i, j). Our electrophysiology data did not indicate larger SV content, though (Fig. 3e, f); possibly, optoSynC clustering within single SVs could alter the TEM appearance of the SV. Whether SNG-1 is part of DCVs, is not known. The size of LVs remained unaffected by photo-stimulation (Supplementary Fig. 5c, k), and the overall number of SVs (+light: 23.89 ± 2.83 nm; -light: 20.53 ± 1.41 nm), as well as the total number of docked SVs (+light: 1.11 ± 0.16 nm; -light: 0.93 ± 0.11 nm) was not significantly increased (Supplementary Fig. 5d, e, l, m). Yet, when we analyzed the distribution of docked SVs relative to the DP (Supplementary Fig. 5f–h), they showed a tendency of being depleted after optoSynC stimulation. However, this excluded those SVs right next to the DP, which were significantly increased after photostimulation (no such effect was observed in *sng-1(ok234)* controls; Supplementary Fig. 5f, g, n, o). Possibly, after illumination and before freezing the samples, SVs that were already docked could fuse, and as SV replenishment from the RP stopped during this time, this caused depletion. At the DP some 'leftover', un-clustered SVs still docked, but then became trapped, instead of diffusing laterally into the AZ membrane[54]. We suggest that optoSynC blocks transmission by impeding SV replenishment from the RP, and that no major structural abnormalities are induced at the PM. However, mobility of SVs along the membrane could become restricted by optoSynC activation.

## optoSynC activation blocks escape behavior in larval zebrafish
We next tested the utility of optoSynC in vertebrates, using zebrafish. Transient expression of a construct comprising synaptophysin, GFP and CRY2olig(535) (termed zf-optoSynC) under control of a 10xUAS element in the pan-neuronal *Tg(elavl3:Gal4-VP16)* driver line[60] was observed in the spinal cord after 24 h and 3 days post fertilization (dpf; Fig. 5a, b). Larvae at 4 dpf were then subjected to swimming assays. In control larvae injected only with eGFP, blue-light exposure (470 nm, 0.6 mW/mm²) caused escape behavior, i.e. an increased rate of swimming speed, which then leveled off (Fig. 5c, d and Supplementary movie 3). In contrast, in larvae expressing the zf-optoSynC construct, no increase in swimming speed could be observed, although they could swim normally before illumination (Fig. 5c, d and Supplementary movie 4). We further assessed neuronal activity of zf-optoSynC expressing larvae using a touch-evoked escape assay, after animals were illuminated for 5 min (470 nm, 0.1 mW/mm²). Control injected larvae responded to touch by escaping, whereas larvae expressing zf-optoSynC showed a significantly reduced response frequency (Fig. 5e). However, they were not unable to show escape, verifying that zf-optoSynC did not per se disturb neuronal activity or behavior. Thus, optoSynC can block synaptic transmission in motor neurons, and/or upstream neurons, to block evoked escape behaviors in zebrafish.

## optoSynC activation attenuates the synaptic vesicle cycle in murine hippocampal neurons
We also tested whether optoSynC functions in mammalian neurons. We expressed a construct of mammalian synaptophysin (SYP), fused to the pH-sensitive fluorescent protein mOrange, without and with

CRY2olig(535) (termed mammalian optoSynC – m-optoSynC) and stimulated synaptic transmission by 40 APs (4 s stimulation at 10 Hz; Fig. 6). As previously reported[61], we observed a transient increase in mOrange fluorescence due to the fusion of SVs and exposure of the SV lumen to physiological pH (thus de-quenching mOrange fluorescence; Fig. 6a, e and Supplementary Fig. 6). Following the increase, fluorescence returned to base line over the next 20 s, due to SV endocytosis and re-acidification. After illumination with a 488 nm LED for 30 s (1 mW/mm²), mOrange2-Syp expressing neurons showed no change in fluorescence increase and recovery with 40 electrical pulses (10 Hz; Fig. 6a–c and Supplementary Fig. 6). However, in m-optoSynC neurons, increase in mOrange fluorescence was significantly attenuated (Fig. 6d–f and Supplementary Fig. 7). These experiments demonstrate that optoSynC can be used also in other organisms and cell systems, and should be a widely applicable tool for synaptic inhibition.

## Cell-type specific inhibition via optoSynC
Thus far, we used pan-neuronal expression of optoSynC. Next, we asked if optoSynC also allows inhibiting synaptic transmission in distinct neuron classes. We expressed optoSynC in subsets of motor neurons, using promoters specific for cholinergic (p*unc-17*, encoding the vesicular acetylcholine transporter; Fig. 7a, b) and GABAergic neurons (p*unc-47*, vesicular GABA transporter; Fig. 7c, d)[62,63]. Expression and activation of optoSynC in cholinergic neurons significantly reduced swimming cycles during light-stimulation (470 nm, 0.1 mW/mm², 5 s, 25 s ISI) which recovered within 15 min after switching off the stimulation (Fig. 7a, b). These effects were similar as for pan-neuronal expression; however, the animals did not slow down as much (reduction by ca. 55%, compared to ca. 80%). Inhibition of GABAergic neurons by optoSynC had less effect, though the reduction of swimming cycles was still significant (Fig. 7c, d). Blocking of GABA neurons recovered similar as in the panneuonal strain ($\tau_{off}$ = 7.87 min). We wondered if simultaneous expression of optoSynC in, and inhibition of, cholinergic and GABAergic neurons would be additive. However, swimming cycles were not any further reduced (Supplementary Fig. 8e, f). Thus, it is likely that the maximal effects we observed by pan-neuronal expression correspond to additive effects of cholinergic motor neurons and upstream pre-motor interneurons. For maximal efficiency, optoSynC should be expressed in *sng-1(ok234)* background, such that the proteins do not compete with endogenous SNG-1 for incorporation into SVs (Supplementary Fig. 8a, d).

## Activated optoSynC blocks transmission from the single nociceptor neuron PVD
Finally, we explored whether optoSynC can affect synaptic transmission in a single neuron pair. To this end, we expressed optoSynC along with Chrimson[64], a red-light activated channelrhodopsin, in the nociceptive PVD neurons[65]. Activation of Chrimson induces rapid forward escape behavior[8] and therefore, a strong increase in crawling speed (Fig. 8a). Concomitant activation of optoSynC may thus cause an attenuation of this escape behavior. We compared animals expressing Chrimson and optoSynC in PVD to animals expressing only Chrimson, before and after applying blue light (5 s, 0.1 mW/mm²), and prior to red light stimulation (680 nm, 0.1 mW/mm², 1 s / 5 s ISI). Though Chrimson is primarily activated by red light, it also shows some response to blue light (Fig. 8d)[64]. Therefore, some pre-activation of Chrimson occurred with the blue light pulse used to trigger optoSynC (Fig. 8a, b). This pre-activation did not affect the consecutive activation by red light (Fig. 8c), and thus the velocity increase of animals expressing optoSynC and Chrimson was significantly inhibited, compared to the controls without optoSynC, or without blue light (Fig. 8c). To optimize this experiment, we tested at which blue light intensities optoSync activation would not evoke activation of PVD::Chrimson (Fig. 8d). This was the case at ≤25 μW/mm². We thus repeated the above experiment, verifying that a single neuron pair can be specifically photoinhibited

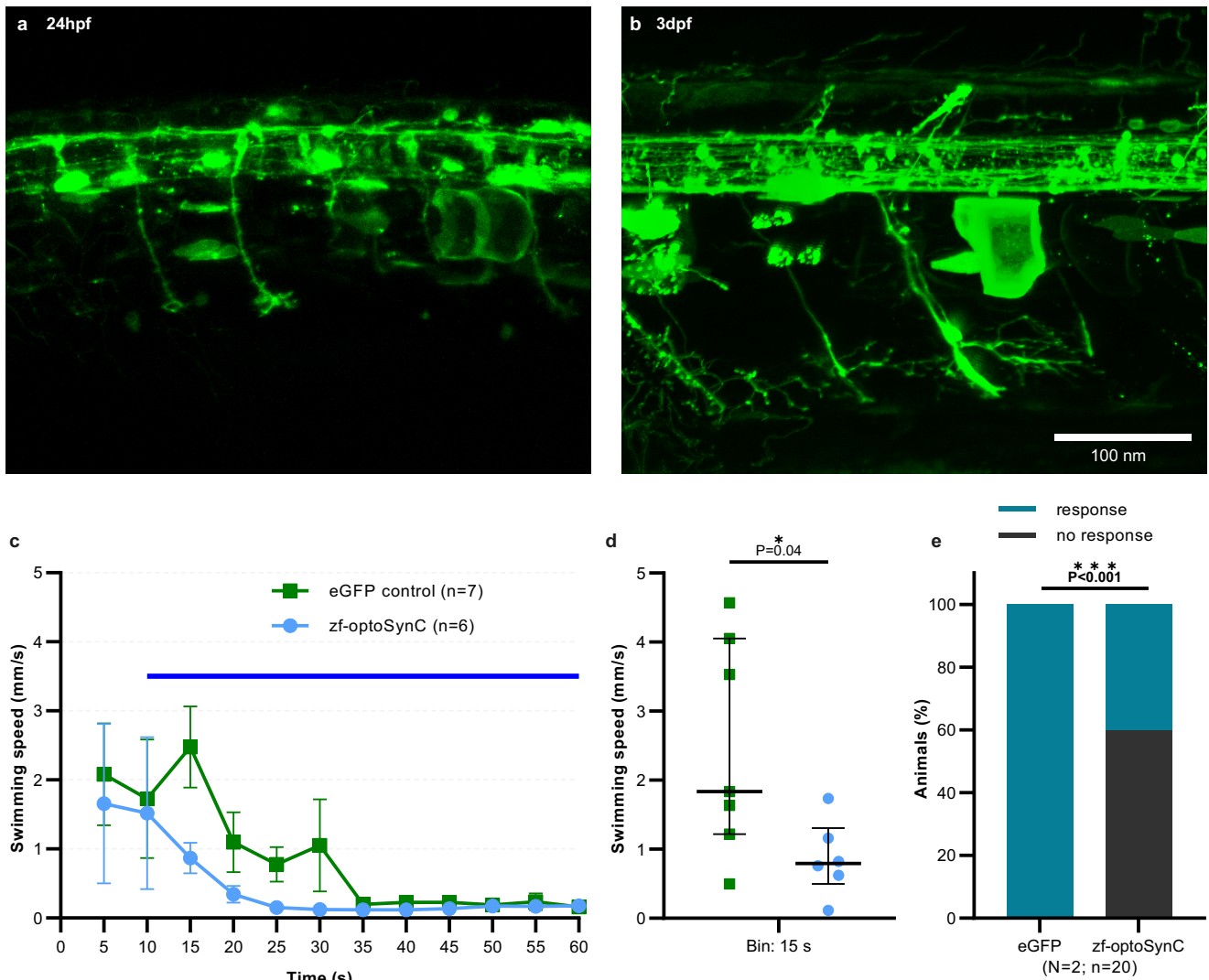

**Fig. 5 | zf-optoSynC activation in zebrafish neurons blocks escape behavior.**
**a**, **b** Pan-neuronal expression of eGFP in neurons of zebrafish larvae at the indicated developmental stages. Representative images show transient expression in *Tg(elavl3.2:Gal4-VP16)mde4* animals injected with UAS:zf-opto-SynC (*n* = 2 independent experiments). **c** Swimming behavior, triggered by continuous blue light (blue bar), in 4 dpf larvae expressing the respective transgene, as indicated. Mean ± s.e.m., numbers of individual animals are indicated. **d** Statistical analysis of data in **c**, as median and 75–25% IQR. Number of independent animals *n* = 7 (eGFP) and *n* = 6 (zf-optoSynC). **e** Touch response of 3 dpf larvae during blue light-induced zf-optoSynC/CRY2olig(535) clustering. Two experiments, number of animals is indicated. Unpaired Student's *t* test (two-tailed) in **d**, and Fisher's exact test (two-sided) in **e**.

using optoSynC (Fig. 8e, f). In sum, optoSynC is a highly sensitive optogenetic tool for spatial, temporal, and cell-specific inhibition of synaptic transmission with fast onset and recovery.

## Discussion

In this work, we present the development of a new optogenetic tool, optoSynC, for light-induced neuronal silencing in vivo. We demonstrate its function in various neuron types of *C. elegans*, in the zebrafish nervous system, as well as in murine hippocampal neurons. CRY2 homo-oligomerization, a modality thus far utilized mainly for induction of signaling pathways by protein–protein interaction[49,66,67], was used to trigger formation of SV clusters, as evidenced by electron microscopy. This way, optoSynC activation efficiently inhibits the release of neurotransmitters within 20 s, and in the dark, the effects recovered within 15 min, as we showed by behavioral, pharmacological, and electrophysiological studies. By specific expression, optoSynC can selectively interfere with activity of neuronal subtypes and even single neuron pairs. optoSynC is a fully reversible optogenetic tool for inhibition of synaptic transmission that does not depend on

membrane currents, metabotropic changes, or induced protein damage. Among such inhibitors, it is, to our knowledge, the fastest available (Fig. 9a).

CRY2 was targeted to the SV by fusion to SNG-1, an integral SV protein of high abundance, which increased the chance of efficient clustering of SVs. SNG-1 is not required for synaptogenesis or neuronal activity[39], and functionally, optoSynC is inert in the dark, as we further showed by mutating the FAD-binding pocket of CRY2[34]. We noted that *sng-1(ok234)* mutants differed from SNG-1::CRY2olig(535) expressing animals in SV distances and diameters (Fig. 4 and Supplementary Fig. 5a). The latter resembled wild type[51,54], thus arguing for rescue of this aspect by optoSynC. By electron microscopy, we found evidence for SVs moving closer together, as the average distances of neighboring SVs decreased by 14%, an effect not observed in the absence of optoSynC. The nearest distance of neighboring SV membranes in the RP, but not at the PM, was reduced by activated optoSynC from around 10 nm to 5 nm. That this was not zero distance may be due to the structural properties and size of CRY2[68] and its oligomers, likely limiting the minimal distance of neighboring SVs. Had SV cluster

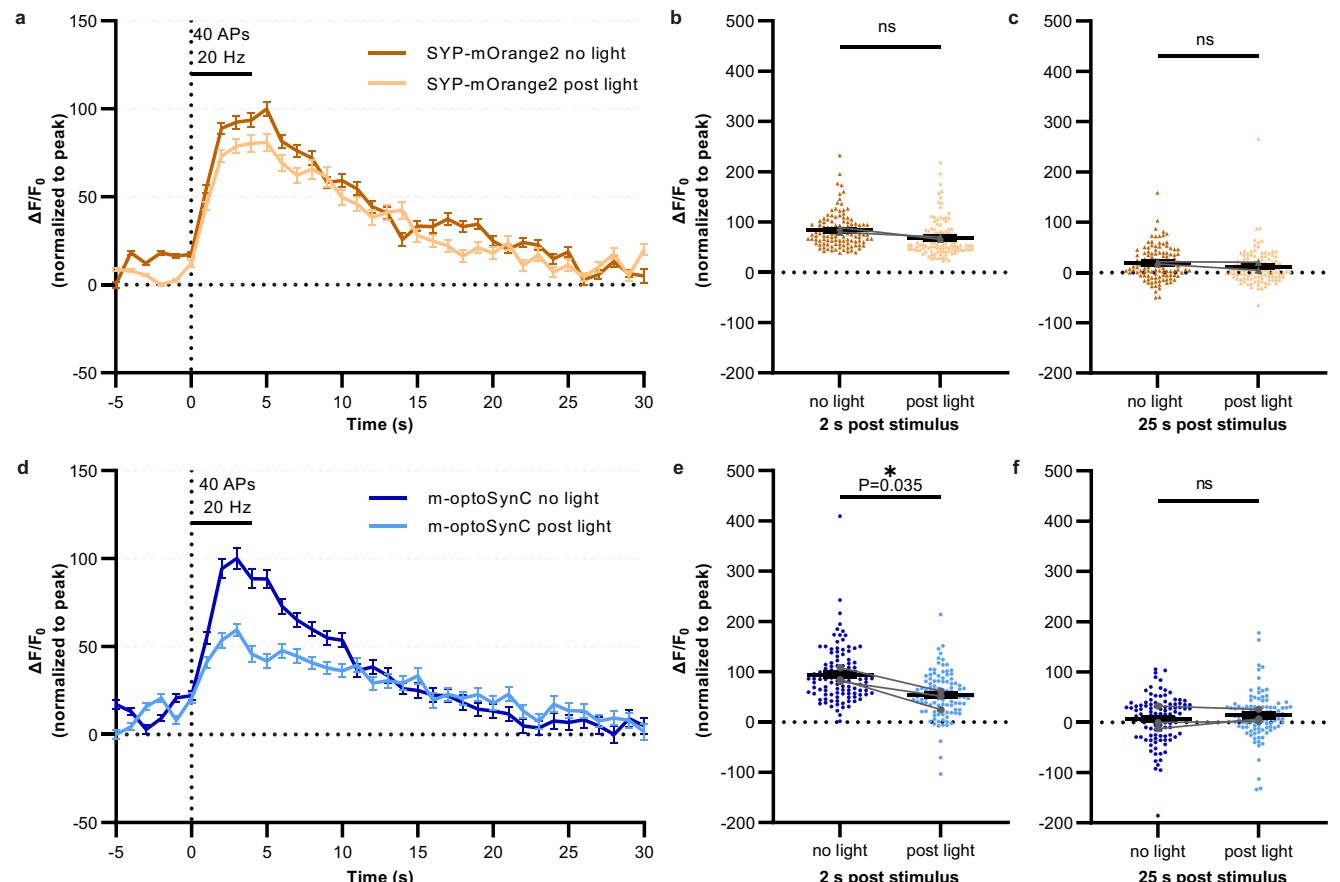

**Fig. 6 | m-optoSynC activation in murine hippocampal neurons blocks synaptic transmission. a** Plot showing changes in normalized fluorescence intensity of mOrange2 by electrical field stimulation of neurons expressing mOrange2-SYP before (black line) and after 488 nm illumination for 30 s (green line). $N = 2$ independent cultures. $n = 5$ neurons and 116 synapses. **b**, **c** Normalized fluorescence intensity at 2 s or 25 s after electrical stimulation. Paired *t*-test. **d**–**f** Same as a-c, respectively, but in neurons expressing m-optoSynC (CRY2olig(535)) inserted into SYP-mOrange2. Paired *t*-test. $N = 3$ independent cultures. $n = 10$ neurons, and 108 synapses. ns = not significant. *<0.05.

formation occurred at the AZ, we would expect abnormal membrane structures, and defective SV recycling should lead to increased formation of LVs[54]. However, no such alterations were observed. We observed an accumulation of docked SVs right at the DP, while other docked SVs were depleted following optoSynC activation. Thus, most likely, SVs in the RP form clusters and cannot be transported to the AZ, thus causing a stop of transmission. Interaction of SVs through CRY2 oligomers seems not to prevent docking. In optogenetically hyperstimulated synapses, we previously observed that SVs became replenished along the PM, and only at late times, also re-appeared at the DP[54]. Thus, newly docked SVs (at the DP) diffuse laterally into the AZ membrane, until this pool is refilled, and only then become observable at the DP in EM snapshots. In optoSynC-stimulated synapses, the few free SVs that remain may dock at the DP, but then be prevented from lateral diffusion, possibly by optoSynC aggregates in the PM.

In zebrafish, panneuronally expressed zf-optoSynC allowed blocking escape behavior. While we cannot conclude where in the escape-circuit(s) the blockade occurred, the experiments demonstrate abolishment of the otherwise evoked behavior. In cultured murine neurons, activation of m-optoSynC effectively blocked synaptic transmission after 30 s illumination. This approach may be compared to the recently described opto-vTrap[38], using a bipartite, LARIAT approach. The m-optoSynC we described here appears to be faster (Fig. 9a), though to firmly conclude this, identical experiments need to be performed side by side.

To characterize the dynamics of optoSynC activation on synaptic transmission, we measured behavioral phenotypes. Activation of optoSynC impaired swimming behavior by 80%, which could be maintained for as long as illumination was applied (several hours, as shown by aldicarb assays). Judging by the inhibition of photophobic behavior, optoSynC may affect transmission already during the first 2–3 s of illumination. By electrophysiology, the maximum effect developed over one minute. Yet, it is unknown how basal activity of the NMJ in dissected animals compares to activity during locomotion in an intact animal, and whether mPSC rate linearly translates into locomotion speed. In intact animals, inhibition of upstream interneuron activity likely increases the effects on behavior. In dissected animals, interneuron - motor neuron connections may have been severed, thus possibly delaying optoSynC effects.

Depending on the assay and on cell-type, optoSynC works with different performance: It mediates neuronal silencing at very low light intensities, swimming is decreased by 80%, mPSC frequency by 60%, and photophobic escape behavior is almost completely blocked. However, when expression was restricted to cholinergic or GABAergic neurons only, its effects on behavior were less (55 and 30 % reduction of swimming speed, respectively). In PVD cells, escape speed was reduced by up to 50%. Thus, effects seem additive if more neurons of a circuit are involved. CRY2 clusters were described to be dynamic[47], possibly explaining remaining neuronal activity during blue light illumination. We recommend using an *sng-1* mutant background, since expression of optoSynC alongside endogenous SNG-1 reduced its

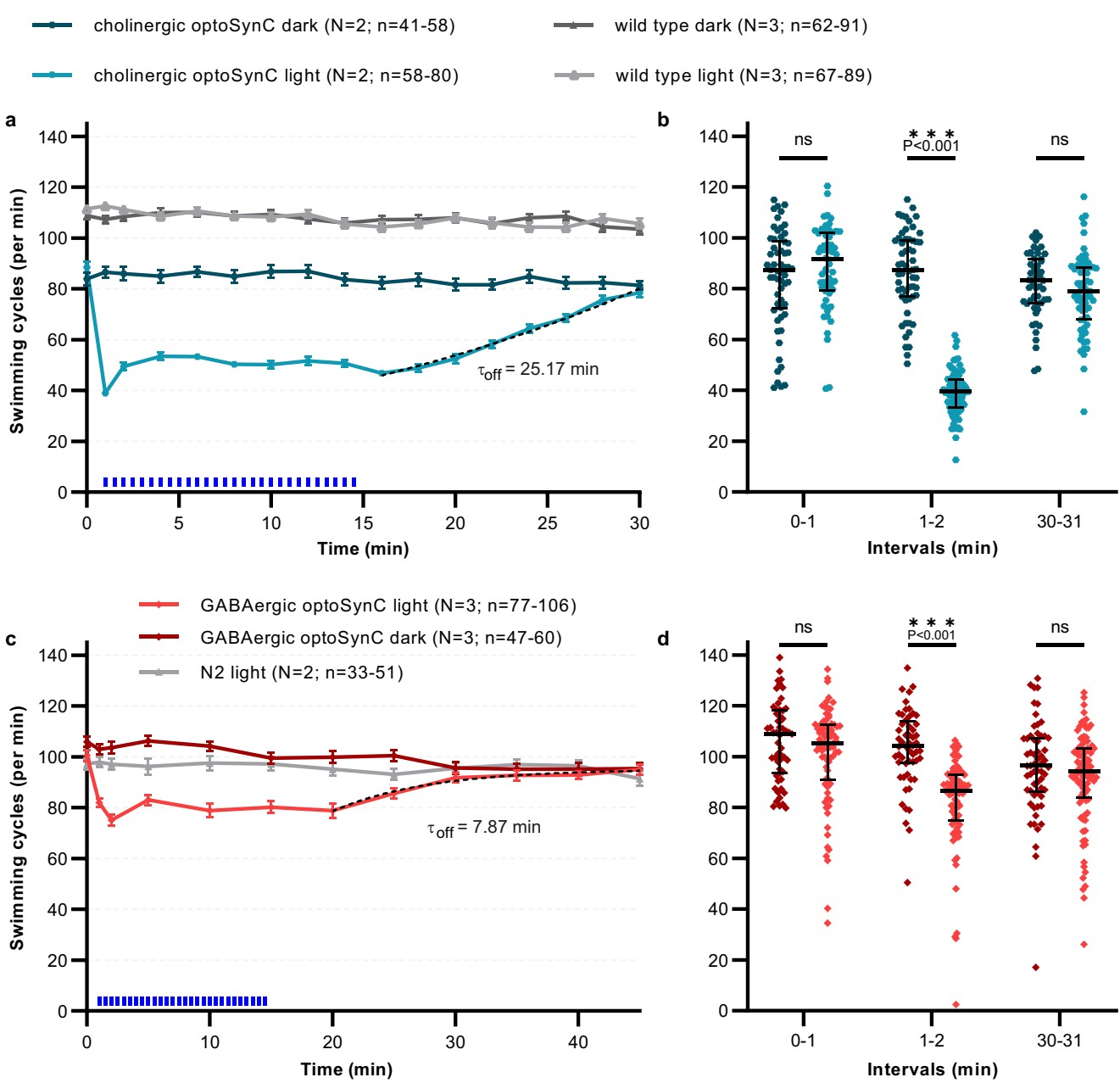

**Fig. 7 | Cell-specific optoSynC inhibition of cholinergic and GABAergic neurons.** **a** Swimming behavior in animals expressing optoSynC in cholinergic neurons. Blue light activation (470 nm, 0.1 mW/mm², 5 s/25 s ISI) indicated by blue rectangles. Dotted line: 'plateau followed by one phase association'-fit. Error bars are s.e.m. Number of individual animals ($n$) across independent experiments ($N$, i.e. animals picked from $N$ independent populations) is indicated as range. **b** Swimming cycles of individual animals, median and 25–75 % IQR of $N = 2$–3

experiments analyzed in time intervals before (0–1 min), during (1–2 min), and after (30–31 min) blue light illumination; number of individual animals ($n$) across all experiments from left to right: 58, 58, 57, 72, 53, 64. Two-way ANOVA with Bonferroni correction between light and dark measurements of wild type and cholinergic optoSynC expressing strains; ***$p < 0.001$. **c, d** as for **a, b**, but optoSynC was expressed in GABAergic neurons. Number of individual animals ($n$) from left to right: 77, 55, 86, 56, 98, 60.

efficacy, likely due to reduced incorporation into SVs. However, this depended on neuron type, and did not affect efficiency in cholinergic neurons. optoSynC is more efficient than other tools designed to inactivate neurotransmitter release by targeting SNARE proteins such as miniSOG/InSynC[27], PSD[28,29], or PA-BoNT[30], particularly with respect to the fast action and recovery (Fig. 9a). These approaches affect protein degradation or inactivation, and are reversible only by de novo synthesis. miniSOG/InSynC[25] reduces swimming behavior of *C. elegans*[28], and can be used in single neurons; however, inhibition takes several minutes to build up and continues after the end of the light pulse. Inducing damaging radicals comes with off-target effects that

can have unknown long-lasting outcomes[27]. PSD and PA-BoNT action (degradation/cleavage of SNT-1) reduce swimming behavior of *C. elegans* by 60–70 % (refs. 28,30), but require almost one hour. In comparison, optoSynC is more efficient, acts immediately, can be sustained for several hours with no known off-target effects, with fast reversibility (15 min compared to 16–24 h[27,30]). optoSynC is much slower compared to light-driven hyperpolarizing ion pumps or anion channels[18], yet as these can inactivate during prolonged illumination[17], optoSynC is advantageous in long-term applications. Light-gated anion channels rely on the Cl⁻-gradient, which can pose problems, e.g. in synaptic terminals[12,20]. When we used ACR2 in cholinergic

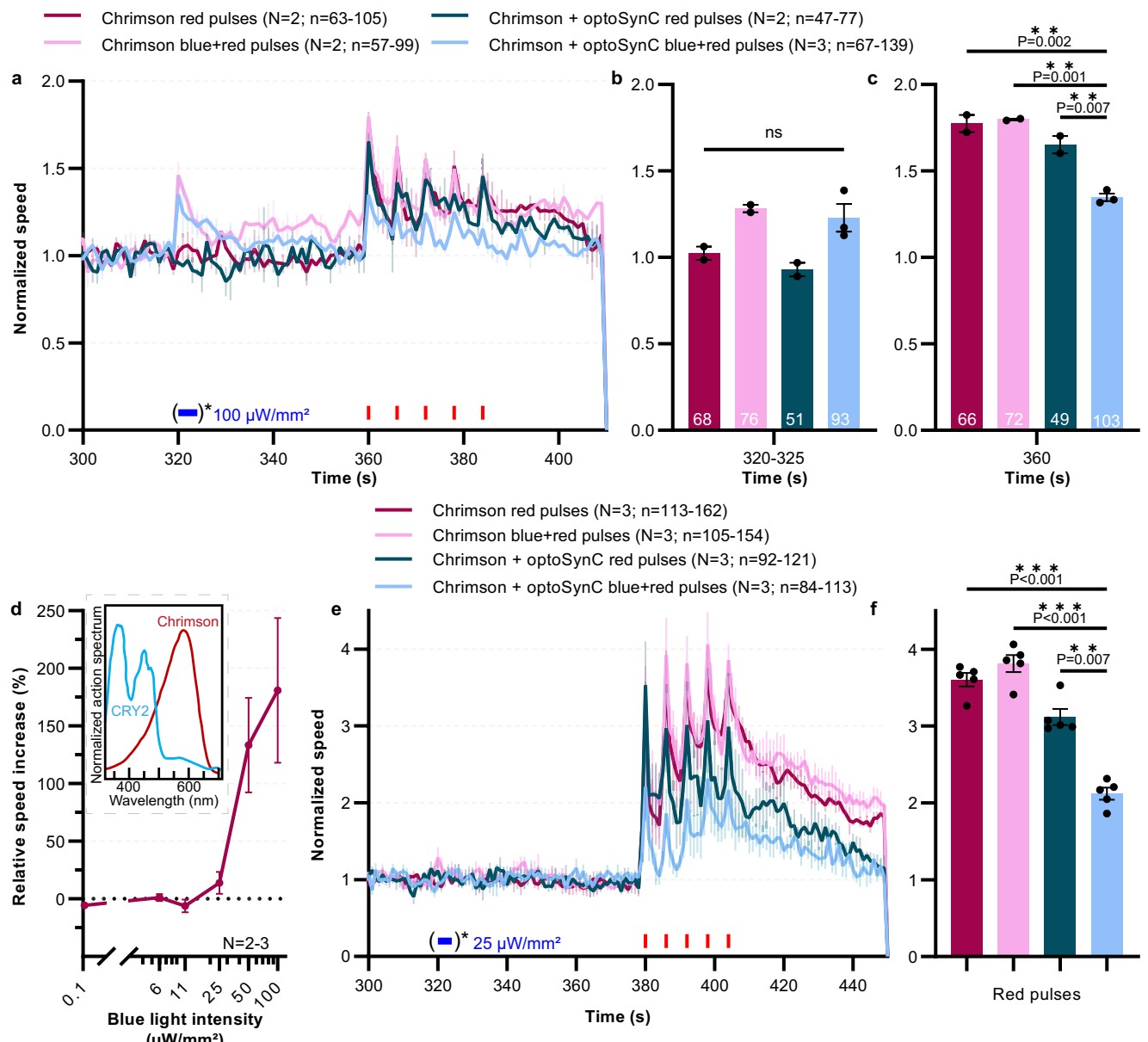

**Fig. 8 | Expression of optoSynC in the single nociceptive neuron PVD attenuates PVD::Chrimson-evoked velocity increase. a** Mean ± s.e.m. normalized crawling speed of animals expressing Chrimson, or Chrimson and optoSynC in the PVD neuron. Red light stimulation (680 nm, 0.1 mW/mm², 1 s/5 s ISI) is indicated by red rectangles, activation of optoSynC with blue light (470 nm, 0.1 mW/mm², 5 s) by a blue rectangle and asterisk. Data acquisition started at 0 s, but animals are left to accommodate before starting the experiment. **b** Group data, mean crawling speed of N = 2–3 independent experiments (±s.e.m) analyzed during time intervals with (or without) blue pulse (320–325 s). Number of independent animals (n) across all independent experiments (N, i.e. animals picked from N independent populations) is indicated as range. **c** As in **b**, but during first red light pulse (360 s). **d** Light intensity titration of blue light responses of PVD::Chrimson animals (mean ± s.e.m.; 61–147 individual animals, across N = 2–3 independent experiments, were analyzed). Inset: Spectral overlap of CRY2 and Chrimson, derived from refs. 44,81. **e** As in **a**, but with only 25 μW/mm² optoSynC activation, avoiding optical crosstalk with Chrimson. **f** as in **c**, but during peaks of red pulses (380 s, 386 s, 392 s, 398 s, 404 s), for data in **e**. In **b**, **c**, **f**: one-way ANOVA with Bonferroni correction; ns non-significant. Number of independent experiments (N) and individual animals (n) is indicated.

neurons for swimming inhibition (30 s; Fig. 9b), a rebound effect occurred, i.e. animals swam faster than before the inhibition, for up to 90 s. Possibly, an aberrant neuronal Cl⁻-gradient, altered during photoactivation, needs to be dissipated before normal locomotion resumes.

The recently developed opto-vTrap[38], based on a LARIAT approach[37], uses heterodimerization of CRY2 with CIB1[33,34]. A similar approach, i.e. expressing soluble CRY2 in synapses, and targeting CIBN to SV membranes via SNG-1, could not reduce synaptic transmission in *C. elegans*. opto-vTrap targets VAMP, therefore blocking SNARE

complex formation is the likely reason for inhibition, while optoSynC clusters SVs. Thus, the tools are complementary; however, optoSynC may additionally enable investigation of different SV pools and individual steps during the SV cycle, as well as dynamics of SV mobilization. While opto-vTrap requires expression of two proteins, optoSynC is functional as a single tool, and as we show here, also works in zebrafish and murine hippocampal neurons. It should thus be transferable to further animal models, and the low light intensities needed for optoSynC may facilitate applications also deep within tissue. optoSynC is a tool for silencing, as well as for studies of the dynamic interplay

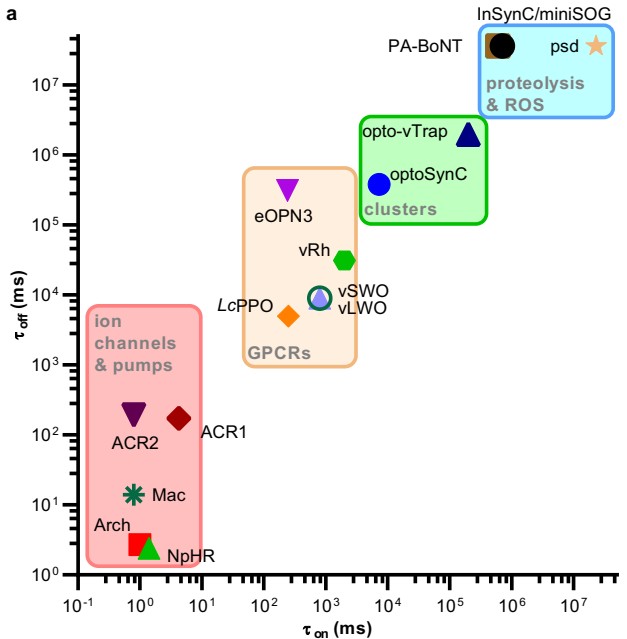

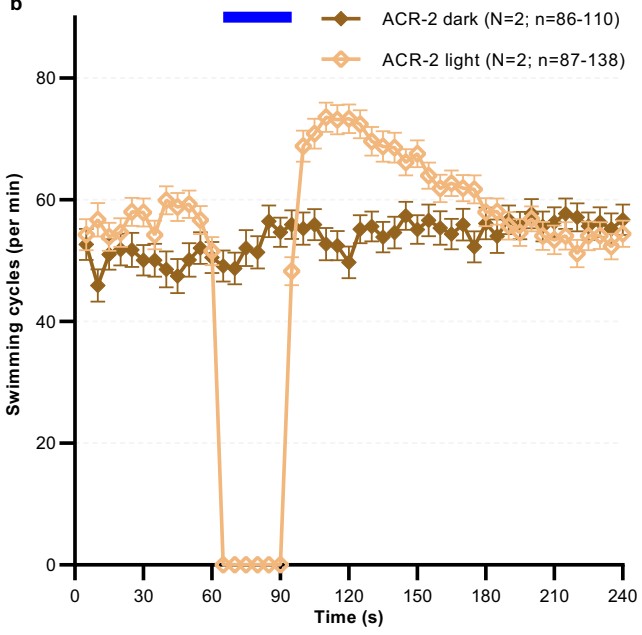

**Fig. 9 | Comparison of tools for optogenetic silencing of synaptic transmission.** **a** Matrix showing time constants (activity on- and offset) of optogenetic tools for synaptic inhibition. Tools are grouped by their molecular nature/mechanism of action. Data was derived from refs. 13,18,19,22–24,27,28,30,38,82,83 **b** ACR2 expressed in *C. elegans* cholinergic motor neurons evokes rapid and complete inhibition of locomotion, associated with sustained after-effects. Mean (±s.e.m.) swimming cycles of animals expressing ACR2. Number of individual animals (*n*) across each measured time point and across *N* = 2 independent experiments (animals picked from *N* independent populations) is indicated as range.

between SV pools. Furthermore, it may allow blocking transmission of neuropeptides or neuromodulators from dense core vesicles. Last, in neurons that release different transmitters, i.e. containing distinct SV neurotransmitter transporters, one type of SVs may be specifically clustered by optoSynC, while others remain unaffected.

## Methods
### Molecular biology
To express optoSynC in *C. elegans*, the promoters p*sng-1* (pan-neuronal), p*unc-17* (cholinergic neurons), p*unc-47* (GABAergic neurons) and *ser2prom* (driving expression in the PVD neuron) were used. As selection marker, we expressed fluorescent proteins under the control of either the promoters p*myo-2* (expression in pharyngeal muscle) or p*myo-3* (expression in body wall muscle cells).

Plasmid **pDV01** [punc-17::CRY2olig(535)] was produced by amplifying the cDNA of CRY2olig(535) using primers oDV01 (5′-TGG CTAGCCGTCGTTCCGGAGGAGGTGGCGCCCGGGATCCAATGAAGATG GACAAAAAGACTA TAGTTTGG-3′) and oMS67 (5′-CTGGGTCGAATT CGCCCTTTCCCTTGTCGACCATGACTCGAGTTAAACAGC CGAAGGTA CTTGTTGG-3′), restriction digest of the fragment using *Sal*I and sub-cloning into vector pAH04 [punc-17::ccb-1::miniSOG::Citrine] by restriction digest with *Msc*I, *Psi*I and *Sal*I. Plasmid **pDV04** [punc-17::mOrange2::CRY2olig(535)] was generated by amplification of mOrange2 sequence with primers oDV013 (5′-CCGCATCTCTT GTTCAAGGGATTGGTGGCTAGGCTAGCCTCGAGAGGCCTGG-3′) and oDV014 (5′-GGGCGCCACCTCCTCCGGAACGACGGCTAGACTTGTAGA GTTCGTCCATTCCTCC-3′) and cloning it into vector pDV01 using *Nhe*I and Gibson Assembly. For **pDV09** [psng-1::mOrange2::CRY2(535)] the p*unc-17* sequence was replaced by the sequence of p*sng-1*, by restriction digest of pMS20 [psng-1::mCherry::BoNTB(N)::iLID] with *Msc*I and *Sph*I and pDV04 with *Nhe*I (blunted with T4 polymerase) and *Sph*I. For **pDV10** [psng-1::SNG-1::eGFP::CIBN], the CIBN fragment was amplified with primers oMS058 (5′-ATACAAAGGGGTTACCGGATCCGGCCTCG AGATGAATGGAGCTATAGGAGGTGAC-3′) and oMS059 (5′- TACGAA TGCTCCCGGGCCTGCAGGCCCTAGGTTAAATATAATCCGTTTTCTCC

AATTCCTTC-3′) and sub-cloning into pMS21 [psng-1::sng-1::eGFP::SspB(milli)_BoNTB(C)] using restriction enzymes *Avr*II and *Psp*XI. **pDV12** [psng-1::SNG-1::CRY2olig(535)] was assembled by amplification of CRY2olig(535) sequence with oMS060 (5′-GA GGGTCCGGTGGCGGAGGGTCAGGGGTACCGATGAAGATGGACAAAA AGACTATAGTTTG-3′) and oMS062 (5′-TACGAATGCTCCCGGGCCT GCAGGCCCTAGGTTATTTGCAACCATTTTTTTCCCAAACTTG-3′) and subcloning into the psng-1::SNG-1 vector gained by restriction digest of pMS21 with *Avr*II and *Kpn*I. **pDV15** [psng-1::SNG-1::eGFP::CRY2olig(535)] was generated by amplifying the CRY2olig(535) fragment with oMS062 (5′- TACGAATGCTCCCGGGCCTGCAGGCCCTAGGTTATTTG-CAACCATTTTTTTCCCAAACTTG-3′) and oMS063 (5′- ATACAAAGGG GTTACCGGATCCGGCCTCGAGATGAAGATGGACAAAAAGACTATAGT TTG-3′) and Gibson assembled with the psng-1::SNG-1::eGFP fragment after restriction digest of pMS21 with *Avr*II and *Psp*XI. The plasmid **pDV06** [punc-17::SNG-1::CRY2(535)] was generated by restriction digest of pDV12 and pMSM17 [5′hom-CCB-1::mKate2::SEC:: cMyc(3x)::iLIDpsd-s::3′hom-CCB-1] with *Kpn*I and *Bsm*I and subsequent ligation. For **pDV18** [punc-47::SNG-1::CRY2olig(535)], p*unc-17* was replaced by amplification of the p*unc-47* promoter with oDV025 (5′-AACAACTTGGAAATGAAATAAGCTTGCATGCCTGCAGAGCTTGTTGT-CAT-3′) and oDV026 (5′-GCACCATAAGCACGCACGTTCTCCATTT CACCGGTGCTGTAATGAAATAAATGTGACGC-3′) and subcloning into restriction-digested pDV06 with *Sph*I-HF and *Sgr*AI-HF. Plasmid **pDV19** [ser2prom3::SNG-1::CRY2olig(535)] was generated by ligation of the restriction digest of pDV06 and pJW41 [ser2prom3::Chrimson::mNeon] with *Bmt*I-HF (blunted with T4 polymerase) and *Sph*I-HF. To construct **pDV20** [pSNG-1::SNG-1::CRY2(D387A)olig(535)], site-directed mutagenesis was conducted using the template pDV12 and the primers oDV027 (5′-ACACTTTTGGCTGCTGATTTGG-3′) and pDV028 (5′-CCAAATCAGCAGCCAAAAGTGT-3′). **pDV21** [pSNG-1::SNG-1::CRY2 (D387A)olig(535)::SL2::mCherry] was generated by ligation of the restriction digests of pDV20 with *Pvu*I-HF and *Sma*I and pTH02 [pmyo-3::CaCyclOp::SL2::mCherry] with *Kpn*I-HF (blunted using T4 polymerase) and *Pvu*I-HF.

For expression of optoSynC in zebrafish larvae, a middle entry clone (pHD53) for Gateway™ cloning was synthesized (Twist Bioscience, San Francisco, USA) using the pTwist ENTR backbone and containing the coding sequence of synaptophysin b (ENSDARG00000002230) fused to *egfp*, as well as a codon optimized version of *CRY2olig(535)*. Subsequently, the resulting middle entry clone was used in an LR-reaction (Thermo Fisher Scientific, Waltham, USA) together with the p5E-UAS, p3E-polyA clones and the pDest-Tol2CG vector[69] to obtain the Tol2-UAS:sypb-egfp-cry2-polyA plasmid (pHD57 / UAS:zf-optoSynC).

## *C. elegans* strains

*C. elegans* strains were cultivated according to standard methods[70] on nematode growth medium (NGM) and fed *Escherichia coli* strain OP50-1. Animals were grown at room temperature and kept in the dark. Transgenic animals were obtained by microinjection of DNA into the gonads of animals in wt background (Bristol N2), *sng-1(ok234)*[39] or *lite-1(ce314)* background[50] by standard procedures[71].

Strains used or generated: Bristol N2, *sng-1(ok234)*, *lite-1(ce314)*, **ZX2483**: zxEx1146[*punc-17::ACR2::eYFP; pmyo-3::QuasAr; pelt-2::GFP*], **ZX2577**: *sng-1(ok234); zxEx1216[psng-1::SNG-1::eGFP::CIBN; pmyo-2::mCherry]*, **ZX2581**: *sng-1(ok234); zxEx1224[psng-1::SNG-1::eGFP::CRY2olig(535); pmyo-2::mCherry]*, **ZX2604**: *sng-1(ok234); zxIs127[psng-1::SNG-1::CRY2olig(535);pmyo-2::mCherry]*, **ZX2628**: *sng-1(ok234); zxEx1234[psng-1::SNG-1::eGFP::CIBN; psng-1::mOrange2::CRY2olig(535); pmyo-2::mCherry]*, **ZX2737**: zxIs132[*punc-17::SNG-1::CRY2(535);pmyo-2::mCherry*], ZX2807: zxIs137[*ser2prom3::Chrimson:: mNeonGreen; pmyo-2::mCherry*], **ZX2816**: zxEx1277[*punc-47::SNG-1::CRY2olig(535); pmyo-3::mCherry*], **ZX2865**: zxIs137; zxEx1291 [*ser2prom3::SNG-1::CRY2olig(535); pmyo-3::mCherry*], **ZX2871**: zxIs132; zxEx1277, **ZX2872**: *lite-1(ce314)*; zxIs127, **ZX2911**: zxIs127, **ZX2914**: *sng-1(ok234)*; zxIs132, **ZX2950**: *sng-1(ok234)*; zxEx1322[*psng-1::SNG-1::CRY2(D387A)olig(535)::SL2::mCherry; pmyo-2::CFP*].

## Zebrafish lines, husbandry, and housing

Adult zebrafish (*Danio rerio*) were maintained in groups in 6 liter tanks (5–7 fish per liter) located in a circulating water system (Zebcare, Nederweert, Netherlands) with a 14 h/10 h light/dark cycle and in accordance with FELASA guidelines[72]. Following microinjection, zebrafish embryos were kept in E3 medium in an incubator at 28 °C and total darkness. Developmental stages of embryos were determined according to landmarks and developmental 'time stamps' as described by Kimmel and colleagues (ref. 73). For all experiments the *Tg(elavl3.2:Gal4-VP16)mde4* strain (European Zebrafish Resource Center, Karlsruhe, Germany) was used. All experiments employing zebrafish were conducted according to the European Directive 2010/63/EU on the protection of animals used for scientific purposes and the animal research board of the State of Hessen.

## Microinjection of zebrafish embryos

Plasmid DNA was diluted to a final concentration of 12.5 ng/µl in water and approximately 0.5 nl were co-injected with in-vitro transcribed Tol2 mRNA (12.5 ng/µl) into 1-cell stage embryos. 2 days post fertilization (dpf) embryos were scored for expression of the cmlc2:EGFP transgenesis marker (Kwan et al. 2007) and used for further experiments.

## Behavioral assays

All strains were kept in the dark on standard NGM plates (6 cm, 8 ml NGM) with OP50-1 bacteria at room temperature. For analysis of swimming behavior, transgenic L4 larvae were selected for fluorescent markers under a Leica MZ16F dissection scope and transferred to freshly seeded NGM plates. After 12–16 h in the dark, 60–80 young adult animals were transferred onto plain NGM plates (3.5 cm diameter, 3 ml NGM) using 800 µl M9 buffer. Animals were kept in

darkness for 15 min for accommodation to swimming behavior. Then, the plate was placed onto a multiworm tracker (MWT) platform[74] equipped with a high-resolution camera (Falcon 4M30, DALSA) and a custom-built infrared transmission light source (6 WEPIR3-S1, WINGER, 850 nm 3 W). Using a LabVIEW-based custom software (MS-Acqu), videos of swimming behavior were automatically recorded for 1 min, at different time points during the experiment. A custom-built LED ring (Alustar 3 W 30°, ledxon, 470 nm) was used for light stimulation controlled by transistor-transistor logic (TTL) pulses from MS-Acqu. Afterward, swimming cycles were automatically analyzed using "wrMTrck" plugin[75] for ImageJ (version 1.52a, National Institute of Health). Tracking was validated, non-worm objects manually removed, worms that were tracked <50% of the total time were omitted, and data summarized using custom Python scripts (https://github.com/dvettkoe/50-percent-tracked). For the acquisition of speed data, the crawling behavior of young adult animals was analyzed on the MWT. Worms were collected in M9 buffer, transferred in a droplet of buffer onto a fresh plain NGM plate (6 cm, 8 ml NGM), and kept for 15 min in the dark. Speed data was recorded using the MWT platform and the respective "Multi-Worm Tracker" software (version 1.3.0)[74]. Tracks were extracted using Choreography.

## Pharmacological assays

To assay aldicarb sensitivity, 2 mM aldicarb plates were prepared by mixing aldicarb (116-06-3, Sigma Aldrich, Germany) from a 100 mM stock solution in 70% ethanol with ca. 50 °C warm NGM, and left to solidify[53]. After cultivation in the dark at room temperature, animals were transferred to the aldicarb dishes (15–27 young adults per trial, three biological replicates in total on 3 consecutive days, animals picked from different populations) and scored every 30 min, for up to 6 h, by three gentle touches with a hair pick to nose and tail regions. All conditions were recorded blinded on the same day. For analysis of the effect of activated optoSynC, animals were illuminated with blue light (470 nm, 0.05 mW/mm²) throughout the 6 h experiment.

## Zebrafish behavioral assays

For behavioral experiments all embryos were kept in darkness, at 28 °C previous to the assay. Plasmid-injected animals were selected for the respective EGFP fluorescence with a Leica MZ16F stereomicroscope the day before the experiment. All assays were conducted at room temperature. For touch evoked response assay, 10 larvae at 3 dpf were placed in a 6 cm dish filled with pre-warmed E3 medium and adapted for 10 minutes under dark conditions. Blue light (470 nm, 0.1 mW/mm²) was applied via a Leica MZ16F stereomicroscope for 5 minutes. Subsequently, larvae were touched with a metal needle at the tail under continuous blue light and movements were scored for a clear escape response by manual inspection. For analysis of swimming behavior, single animals at 4 dpf were transferred to an agarose arena with a diameter of 3 cm filled with pre-warmed E3 medium and adapted for 2 minutes under dark conditions. Videos were taken at 30 fps, for 60 seconds. After a dark phase of 10 seconds, continuous blue light (470 nm, 0.6 mW/mm²) was applied from an LED ring for 50 seconds. X- and Y-coordinates of zebrafish larvae were identified for each video frame in ImageJ, using step-by-step procedures described by Creton[76] and swimming speed was calculated in Microsoft Excel.

## Electrophysiology

Electrophysiological recordings from body wall muscle cells were done in dissected adult worms as described[77]. Briefly, animals were immobilized with Histoacryl glue (B. Braun Surgical, Spain) and a lateral incision was made to access NMJs along the anterior ventral nerve cord. Removal of the basement membrane overlying body wall muscles was enzymatically achieved by incubation in 0.5 mg/ml collagenase for 10 s (C5138, Sigma Aldrich, Germany). The integrity of body wall muscle cells and nerve cord was examined via DIC

microscopy. Recordings from BWMs were acquired in whole-cell patch-clamp mode at 20–22 °C using an EPC-10 amplifier equipped with Patchmaster software (HEKA, Germany). The head stage was connected to a standard HEKA pipette holder for fire-polished borosilicate pipettes (1B100F-4, Worcester Polytechnic Institute, USA) of 4–10 MΩ resistance. The extracellular bath solution consisted of 150 mM NaCl, 5 mM KCl, 5 mM $CaCl_2$, 1 mM $MgCl_2$, 10 mM glucose, 5 mM sucrose, and 15 mM HEPES, pH 7.3, with NaOH, ~330 mOsm. The internal/patch pipette solution consisted of K-gluconate 115 mM, KCl 25 mM, $CaCl_2$ 0.1 mM, $MgCl_2$ 5 mM, BAPTA 1 mM, HEPES 10 mM, $Na_2ATP$ 5 mM, $Na_2GTP$ 0.5 mM, cAMP 0.5 mM, and cGMP 0.5 mM, pH 7.2, with KOH, ~320 mOsm. Activation of light was performed using a LED lamp (470 nm, 8 mW/mm²; KSL-70, Rapp OptoElectronic, Germany), controlled by an EPC-10 amplifier and Patchmaster software (HEKA, Germany). Samples were illuminated using a 5 s/5 s ISI for 30 s. Analysis of mPSCs was done using 'Mini Analysis' software (Synaptosoft, Decatur, GA, USA, version 6.0.7). Amplitude and mean number of mPSC events per second were analyzed during the following time bins: 30 s before illumination and 20 s to 30 s after illumination (Origin Pro 2021).

## Confocal laser scanning microscopy
For imaging purposes larvae were treated with 0.2 mM 1-phenyl-2-thiourea (Sigma-Aldrich, St. Louis, USA) from 24 hpf on to prevent pigmentation. In-vivo imaging was performed on larvae embedded in 1.5 % low melting point agarose (Thermo Fisher Scientific) in E3 medium supplemented with 4.2 g/l MS-222 (Sigma-Aldrich). Confocal imaging was done with a Zeiss LSM 780 microscope using a Plan-Apochromat 20x/0.8 M27 objective. Images were processed with ImageJ.

## Transgenesis, fluorescence imaging and stimulation of murine hippocampal neurons in culture
For lentivirus production, HEK293T cells were maintained in Dulbecco's Modified Eagle's Medium (DMEM) supplemented with 10% FBS (fetal bovine serum) and 0.2% penicillin-streptomycin. One day after plating $6.5 \times 10^6$ cells in T75 flask (Corning), medium was replaced with Neurobasal-A media supplemented with 2 mM GlutaMax, 2% B27 and 0.2% penicillin-streptomycin. Cells were transfected using polyethylenimine with a pFUGW plasmid encoding Syp-mOrange2 with or without CRY2olig(535), and two helper plasmids (pHR-CMV8.2 deltaR and pCMV-VSVG) at a 4:3:2 molar ratio. Three days after transfection, supernatant was collected and concentrated to 20-fold using Amicon Ultra-15 10 K centrifuge filter (Millipore).

All the mouse experiments were performed in accordance with rules and regulations of the National Institute of Health, USA, and animal protocols were approved by committee of animal care, use of the Johns Hopkins University.

To monitor the synaptic vesicle cycle in mouse hippocampal neurons, mOrange2, was inserted in the luminal domain of SYP. For m-optoSynC, CRY2olig(535) was additionally attached to the C-terminus of this construct. Lentiviruses carrying these constructs were generated as described above. Mouse hippocampal cultures were prepared as previously described[78,79]. Briefly, embryonic day 18 (E18) C57/BL6-N mice of both genders were decapitated. The brains were dissected from these animals and placed on ice cold dissection medium (1 x HBSS, 1 mM sodium pyruvate, 10 mM HEPES, 30 mM glucose, and 1% penicillin-streptomycin). Hippocampi were dissected under a binocular microscope and digested with papain (0.5 mg/ml) and DNase (0.01%) in the dissection medium for 25 min at 37 °C. After trituration, dissociated hippocampal neurons were seeded on 18-mm coverslips coated with poly-L-lysine (1 mg/ml) in 0.1 M Tris-HCl (pH 8.5) at a density of $25–40 \times 10^3$ cells/cm². Neurons were cultured in Neurobasal media (Gibco) supplemented with 2 mM GlutaMax, 2% B27, 5% horse serum and 1% penicillin-streptomycin at 37 °C in 5% $CO_2$. Next day, medium was switched to Neurobasal with 2 mM GlutaMax and 2% B27 (NM0), and neurons maintained thereafter in this medium. Half of the medium was refreshed every week. Neurons were infected at DIV (days in vitro) 3 by adding viruses to each well of neurons with 8 μl per well (12-well plate 125k cells per well). Cells were cultured with virus until DIV 13–15 and imaged on a Nikon Ti2E, equipped with a Hamamatsu ORCA-FusionBT and Lumencor Spectra III. Coverglasses were imaged with a ×40 objective lens (NA 1.3, oil immersion) at 37 °C, while the physiological saline solution (140 mM NaCl, 2.4 mM KCl, 10 mM HEPES, 10 mM Glucose, pH 7.3, 300 mOsm, 4 mM $CaCl_2$, and 1 mM $MgCl_2$) was constantly perfused into the chamber. 30 μM bicuculline and 3 μM NBQX were added to the solution to block the recurrent network activity during stimulation. Neuron activation was performed with extracellular field stimulation. Solutions were delivered through a custom heated flow-pipe. Electrical stimulation was applied at the 5th frame for 4 s at 10 Hz (40 APs). NIS-Elements AR software were used for the image acquisition. Images were collected with 100 ms exposure every second, with the first 5 frames collected before field stimulation and the last 30 frames being collected after stimulation for a total of 35 s. Images were acquired using 555 nm excitation LED. Light-induced clustering was achieved using 488 nm light at 1 mW/mm² for 30 s with light continuously on. The same regions-of-interest were imaged 5 s after the 488-nm light exposure. ImageJ was used for all image analysis. Regions of interest were selected based on changes in fluorescence intensity in images collected prior to 488-nm exposure by circular ROIs. Raw fluorescence intensity was measured and plotted in GraphPad-Prism (version 8.02 or version 9) and baseline corrected (fractional difference) to the first 5 frames and then normalized to the peak being 100%. Data from the timepoints of 2 and 25 s post electrical stimulation were used for comparisons at peak exocytic activity and recovery, respectively.

## Electron microscopy
High-pressure freezing (HPF) fixation of young animals was performed as described earlier[54,80]. In brief, 20–40 worms were transferred into a 100 μm deep aluminum planchette (Microscopy Services) filled with E. coli OP50, covered by a 0.16 mm sapphire disk and a 0.4 mm spacer ring (Engineering office M. Wohlwend) for photostimulation. To prevent the preactivation of optoSynC, handling of worms was performed under red light. Animals were illuminated with a 470 nm blue LED (0.1 mW/mm²) for 5 s followed by high-pressure freezing after 25 s at −180 °C under 2100 bar pressure in an HPM100 machine (Leica Microsystems). After freezing, samples were transferred under liquid nitrogen into a Reichert AFS machine (Leica Microsystems) for freeze-substitution. Samples were incubated in tannic acid (0.1% in dry acetone) fixative at −90 °C for 100 h. This was followed by a process of washing to substitute with acetone and incubation in 2% $OsO_4$ for 39.5 h (in dry acetone) while slowly increasing the temperature up to room temperature. Afterwards, an epoxy resin (Agar Scientific, AGAR 100 Premix kit hard) embedding process was performed with increasing concentration from 50% to 90% at room temperature and 100% at 60 °C over 48 h. Cross-sections of 2 individual animals, and from 9–30 synapses, per treatment were cut at a thickness of 40 nm, transferred on formvar-covered copper slot grids, and counterstained with 2.5% aqueous uranyl acetate for 4 min, followed by washing steps with distilled water. Then, grids were incubated in Reynolds lead citrate solution for 2 min in a $CO_2$-free chamber and washed again with distilled water. The ventral nerve cord region was imaged using a Zeiss 900 TEM, operated at 80 kV, and a Troendle 2K camera. For analysis of images, plasma membrane (PM), dense projection (DP), synaptic vesicles (SVs), docked SVs, dense core vesicles (DCVs) and large vesicles (LVs) were annotated in ImageJ (version 1.53c, National Institute of Health) using the synapsEM workflow[58]. Annotated images were analyzed using affiliated MATLAB (R2021a, MathWorks) scripts and a custom script calculating nearest vesicle distances (https://github.com/shigekiwatanabe/SynapsEM).

## Data and statistical analysis

Data are shown as mean ± s.e.m. or as median with interquartile range, $n$ indicates the number of animals, and $N$ the number of biological replicates. Significance between datasets after one- or two-way ANOVA with Bonferroni's multiple comparison test is given as a $p$-value. If data without normal distribution were compared, we used Mann–Whitney's test. Comparison between relative frequency distributions was compared with Kolmogorov-Smirnov test. The respective statistics used are indicated in the figure legends. Data were analyzed and plotted in GraphPad Prism (GraphPad Software, version 8.02).

## Reporting summary

Further information on research design is available in the Nature Portfolio Reporting Summary linked to this article.

## Data availability

Data used to generate the analyses and statistics are provided in a supplementary file "source data". Videos from which these data were generated, as well as fluorescence or electron micrographs used, are available from the authors on request. Source data are provided with this paper.

## Code availability

Custom written scripts for processing of swimming assays are available on GitHub (https://github.com/dvettkoe/50-percent-tracked). The Labview-based script "MS-Acqu" to capture videos for swimming assays using the multiworm tracker platform[74] was hardware-coded, however it will be made available from the authors on request. Scripts calculating nearest vesicle distances following synapsEM workflow[58] are available on GitHub (https://github.com/shigekiwatanabe/SynapsEM).

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

## Acknowledgements

We thank Chandra Tucker for providing CRY2 plasmids. We thank members of the Gottschalk lab for critical comments. We acknowledge the *Caenorhabditis* Genetic Center (CGC), which is funded by NIH Office of Research Infrastructure Programs (P40 OD010440), and the National Bioresource project, nematode *C. elegans*, for strains. We are indebted to Franziska Baumbach, Hans-Werner Müller, Barbara Janosi, Marion Basoglu, and Marius Seidenthal for expert technical assistance. This work was supported by grants from the Deutsche For-schungsgemeinschaft (DFG), SPP 1926, Project VIb (GO1011/12-2), GO1011/19-1, CRC1080 (Project B2), and by Goethe University Frankfurt to A.G.; S.W. is supported by National Institutes of Health (1DP2 NS111133-01 and 1R01 NS105810-01A1. S.W. is an Alfred P. Sloan fellow, McKnight Foundation Scholar, and Klingenstein and Simons Foundation scholar. B.D.G. was supported by the National Science Foundation Graduate research fellowship program (grant # DGE-2139757).

## Author contributions

This work was conceptualized by M.S., D.V., H.D., S.W., and A.G. The methodological custom software used in this work was written by S.W., M.S., and D.V. Data acquisition was done by D.V., M.S., B.D.G., H.D., J.F.L., S.Z., J.G., and Y.A.A. Data was analyzed by D.V., B.D.G., H.D., J.F.L., and Y.A.A. Visualization of data in this work was performed by D.V., B.D.G., H.D., and A.G. Funding was acquired by S.W. and A.G., and the project was administered and supervised by S.W. and A.G. The original draft of this publication was written by D.V. The final draft was reviewed and edited by D.V., H.D., S.W., J.F.L., J.G., and A.G.

## Funding

## Competing interests

The authors declare no competing interests.
