## [Peer Review File · Nature Communications]

Rapid and reversible optogenetic silencing of synaptic transmission by clustering of synaptic vesiclesREVIEWER COMMENTS

Reviewer #1 (Remarks to the Author):

In their manuscript, Vettkötter et al. describe the development and application of a new optogenetic tool "optoSynC" that enables neuronal inhibition via light induced clustering of synaptic vesicles. The method relies on homooligomerization of AtCRY2, a blue light photoreceptor, targeted to vesicles using fusion to synaptogyrin (SNG-1). The authors proof the functionality by various methods in the nematode *C. elegans*, namely by behavioral readouts, electrophysiology, TEM imaging and further demonstrate applicability in different neuronal cell types as well as in a spectral multiplexing experiment with the red light activated channelrhodopsin Chrimson. The carried out experiments appear well performed and always include appropriate controls. However, as the authors have notified and discussed, a similar approach has been described recently (Won, J. et al. Opto-vTrap, an optogenetic trap for reversible inhibition of vesicular release, synaptic transmission, and behavior. *Neuron* 110. 2022). Interestingly, the Opto-vTrap resembling construct was not successful in *C. elegans*. The approach presented by Vettkötter et al. is somewhat more simplistic (homooligomerization versus heterooligomerization) and the application is so far limited to *C. elegans* only and would benefit from application in other animal systems or proof of concept experiments in mammalian neurons. Further, the authors point out the superiority of optoSynC for presynaptic inhibition, mainly by referring to kinetic properties and reversibility, disproportionately and at multiple passages in their manuscript (detailed below). If comparisons with other tools for presynaptic optogenetic inhibition are desired, they should be performed thoroughly and correctly. As optoSynC appears to be a versatile and promising tool for presynaptic inhibition, the manuscript should be suitable for publications after revision and addressing the comments detailed below.

1) Title, Abstract, lines 65-68, 96-97, 182, 382-38 and others: In context of the title as well as the entire manuscript, the authors should be very careful how to discuss and compare temporal terms (fast, rapid, temporal precise, instantaneously, etc.) regarding OptoSynC itself and in comparison with other optogenetic tools. It is further not clear what the authors are referring to when stating that optoSynC is the fastest tool that does not depend on hyperpolarization. Although GPCR-mediated inhibition can activate K⁺-channels, the mechanism of synaptic inhibition is independent of K⁺-channel activation. While they judge that GPCR activation has slow onset "dozens of ms to sec", they describe their tool as "rapid" although the on kinetic is about 15s in the presented behavioral assay. In the current form of the manuscript the kinetics appear to be heavily oversold at multiple passages, which should be revised in a careful and correct manner. Further the authors should also cite at least the recent literature regarding optogenetic GPCR-mediated synaptic inhibition (e.g. Mahn et al. and Coptis et al., both *Neuron* 2021).

2) optoSynC appears to be a potentially useful optogenetic tool for modulation of the presynapse. However the presented data shows only application in *C. elegans*. For the broad neuroscience community using optogenetic tools, the manuscript would highly benefit from an application in mammalian neurons. I am aware that this would go beyond the expertise of the Gottschalk lab and therefore listed this as a minor suggestion, but I would still like to emphasize that I would highly recommend it.

3) Data representation:

a) Much data in this manuscript is presented as bar graphs showing only mean +- SEM. Single point data should be provided wherever possible to allow the reader to estimate the data distribution. Violin or box plots could be used as an alternative or even better in addition to the current data representation. Especially for $n < 10$ (mPSC data) single data points should be shown.

b) X-axis breaks: The authors show initial velocities, followed by axis breaks until the relevant part of their experiments starts/ worms show constant behavior. I think that the initial velocities are not

necessary to show here and that the axis could start right from 250/300s of the experiment. The authors can provide the data starting from time point zero at the source data.

c) Y-axis breaks: are not appropriate as they distort the strengths of the observed effect.

d) Wherever paired data is shown it preferably should also be presented as such.

4) Figure 3d and f: In the statistics of the mPSC frequency analysis the authors binned data for 30s prior optoSynC activation but only for 10s post blue light activation. The analyzed time frames should be equal. (This should ideally be the case for all statistics / binning done within the manuscript unless there is a qualifying reason not to do so).

5) Statistics & Data:

a) In several plots p-values (asterix) as well as the same statistical method are defined multiple times within one figure legend. This should be summarized into one definition per figure where possible.

b) Test statistics (F, t, r) and exact p values are not reported.

c) Source data should be reported.

6) Spectral multiplexing experiment in combination with Chrimson: While in its current form the experiment reveals the fast onset of the clustering based inhibition it also demonstrates optical crosstalk between the blue light activated optoSynC and the red light activated Chrimson. The latter is known to be also activated by blue light as also demonstrated by the authors. However, given the high light sensitivity and long-lasting activity of optoSynC the authors should be able to demonstrate crosstalk-free low intensity blue light activation of optoSynC without activating Chrimson. Ideally the Authors could combine their optoSynC light sensitivity data with measurements of Chrimson activity to calibrate for a final experiment demonstrating the above mentioned crosstalk-free synaptic inhibition with upstream Chrimson activation with red light.

7) The authors should provide a light titration curve and calculate the EC50 of optoSynC from the data that they already have reported in the supplemental material. Ideally, this should be done in comparison to Chrimson activation with blue light as mentioned under 6). The authors could also include a subpanel with the CRY and Chrimson activation/absorption spectra to visualize the source of the potential optical crosstalk. Further, the authors should directly compare the pulsed vs. continuous illumination in the above mentioned sensitivity graph. This information is very valuable for the reader and should not be hiding in the supplement.

8) Methods section: "mm" should be replaced by "mM" in the electrophysiology part.

7) The authors have demonstrated reversibility of optoSynC. For some experiments it would be of interest to show the ability to repetitively perform presynaptic vesicle clustering.

Sincerely, Jonas Wietek

Reviewer #2 (Remarks to the Author):

The manuscript by Vettkotter et al. introduces optoSynC, a new optogenetic tool to suppress neuronal activity. The presented strategy is based on photoinduced homo-oligomerization of cryptochrome-2 (CRY2), which in turn is fused to the synaptic vesicle (SV) protein Synaptogyrin (SNG-1). This setting enables light-triggered SV clustering, resulting in reduced neurotransmitter release from the presynaptic active zone and decreased neuronal signalling. As proof-of-principle, the authors carry out electrophysiology, high-pressure freezing electron microscopy, and behavioural analyses in *C. elegans*.

This nicely designed innovative study has been performed elegantly using state-of-the-art

technologies. The detailed analyses support the authors' conclusions and I anticipate that the manuscript will attract considerable attention extending beyond the field of *C. elegans* Neuroscience. My main concern relates to the utility of the newly developed optoSynC approach. A main advantage of optogenetic strategies is the ability to functionally target specific neuronal subpopulations. However, the cell-specific expression of optoSynC appears to provide only approx. 50% activity suppression, as judged by the locomotion assays in Fig. 5a. Similarly, the electrophysiological measurements of mEPSCs at the single cell/muscle level also show only a roughly 50% reduction in frequency. This moderate inhibition, combined with the necessity to express the CRY2-based fusion construct in the *sng-1* mutant background, will likely limit many applications. I fully agree with the authors' explanation of the advantages optoSynC offers over tools such as miniSOG or PA-BoNT. However, for the interested user it would be important to illustrate in more detail why, at least for certain applications, optoSynC could be preferable over light-gated ion channels, which offer tighter temporal control. The authors describe the potential issues associated with variable chloride concentrations and the danger of in fact activating instead of inhibiting cellular processes by photoinduced chloride conductances. I feel the current study would be significantly strengthened by supporting this argument with a direct comparison between an ACR and optoSynC in an experimental setting.

Other than that, I have only minor comments:

- Line 24: "causing approximation". Consider rephrasing for the sake of clarity.
- Figure 1c, d: following photoactivation of optoSnyC does swimming fully recover at later timepoints (>30 min)?
- Line 178: "of blue light", delete "of".
- Figure 4: Upon light-induced SV clustering the mean nearest SV distances drop to approx. 40 nm. Does this distance match the calculated length of the SNG-1-CRY2olig link?
- Figure 6a: consider denoting the light-stimulated conditions (pink and light blue with asterisks) more clearly.
- Could the authors explain or speculate why the LARIAT approach does not work for them?
- Why do the authors in some cases refer to CIB1 and in other cases to CIBN?

Reviewer #3 (Remarks to the Author):

In this study, the authors report a novel and a much needed optogenetic approach for rapid and reversible silencing of target neurons without affecting neural polarization. The method relies on the light-induced oligomerization of cryptochrome CRY2 which leads to clustering and sequestration of synaptic vesicles. While the silencing is limited in its effectiveness (e.g., some analyses showed less than 50% inhibitory effects), this method still offers a unique and valuable approach that may be employed in many future studies. Furthermore, while the current study demonstrated the applicability of the method in *C. elegans* neurons only, this method could in principle be adapted for use in other model animals.

There are several major concerns that the authors should address:

1. Figure 1c: it seems that the optoSynC-expressing animals do not fully recover by the 30 min of experimental data shown in the figure. Do they really eventually fully recover? Could you show the data after the 30 minutes?

Similarly, in figure 5c: do the GABAergic neurons fully recover? The speed decrease is very low but is it possible to show longer time periods to demonstrate the full recovery?

2. Figure 2a: It is not clear why the speed of optoSynC animals (wt, cyan and *lite-1* bkg, yellow) does not decrease upon light stimulation, just as was noted for the swimming behavior. The speed seems to remain ~constant, arguably merely inhibiting the expected speed increase upon exposure to the blue light. However, I would expect the speed to considerably drop down as shown in figure 1. Could it

be that this strain is defective speeding up upon exposure to blue light (despite the fact that optoSynC-DA can speed up)? Perhaps it is possible to use the optoSynC in the lite-1 background and induce a speed up by poking the tail with and w/o light to observe the slow down.

3. Figure 3g: To explain the results, the authors suggest that optoSynC inhibits ACh secretion and that is why they are slow to get paralyzed. If this indeed the case, then it is mandatory to show the speed by which the animals moved during these hours (for all strains and conditions). It is postulated that optoSynC animals moved significantly slower than the other animal groups throughout the experimental time frame.

Moreover, continuous blue light for such long durations may affect the animals and their locomotive behavior. It is therefore important to repeat the experiment with light on and in the absence of Aldicarb to assess the effect of light itself.

4. Figure 4: Analysis of the EM data was used to infer a possible mechanism for the inhibited SV release. However, an important control is missing: The small difference observed in the distances between the SVs could also be attributed to the exposure to blue light that is known to trigger neural activity. It is appreciated that repeating the experiments to compare WT animals in light and dark is a tremendous effort, but could the authors at least provide an explanation to refute such possibility. This is important since if light itself can explain the minute differences, then the entire logic that strives to explain the inhibitory mechanism may collapse.

5. For many of the comparisons in figure 4 and the relevant supplementary figure S3, the difference between – and + light is minute (e.g 2 nm difference on a scale of >30 nm distances). Despite the fact that such differences come out as statistically significant, it looks like this is due to the large sample size (e.g fig 4c and suppl. fig 4a-b). For example, suppl. fig 4c does not come out significant presumably due to the much smaller sample size. Since the mechanism underlying the inhibition relies on such significance differences, its plausibility is questionable.

Also, the Mann-Whitney test was used to extract significance. However, this test assumes that the variables between and within the groups are independent. So if EM micrographs were from the same animal, or if different synapses were analyzed from the same neuron, then the extracted variables may not be considered independent. The methods section describes that 20-40 animals were used for each group, but could it be that most of the EM data were eventually collected from a few worms?

6. Figure 6a: again, I do not understand why there is no reduction in the basal speed of the animals following exposure to light.

7. Since optoSynC is sensitive to very low intensities of blue light, is it also sensitive to ambient laboratory-typical light? Were such controls conducted? For example, ideally, should all handling with these transgenic animals be done in the dark?

8. Overall, this new tool seems to provide only partial inhibition. For example, ~50% reduction in mPSC frequency. 50% and 20% reduction in the swimming cycles when expressed in cholinergic and GABAergic neurons, respectively. With this caveat, the tool may not be suitable to various applications. Will be good if the authors acknowledge that and discuss which experimental designs may benefit using this tool at its current fair efficiency.

Minor issues to be addressed:

1. The data presented in Supple fig 1 is presumably using a pan neuronal expression but it is not indicated in the text nor in the legend.
2. Suppl. fig. 2c. I believe the x-axis units are minutes (not sec).
3. In line 269, values in the parentheses are provided as a range but it is not specified what is this range. This is just an example as such value ranges repeat throughout the text.
4. the one-phase decay fit does not seem to be accurate. It is evident from supplementary figure S2d

that while the two decaying curves are very similar, the resulting tau values are very different. This is because the decay function actually proceeds past the relevant time period shown in the graph. To extract more accurate tau values, perhaps the authors should consider to include additional data points that are beyond the specified time window. The same holds for the tau computed in figure 1b.

5. Supplementary figure S2b: is there an explanation why animals expressing the GFP-tagged version of the protein show increased swimming cycles after being exposed to blue light?
6. In supplementary figure 4, maybe instead of showing the absolute values of DVs, it would be better to show the ratio between docked and undocked vesicles?

Response to the reviewer's comments and our additions and adjustments.

** Please replace your bar graphs with plots that feature information about the distribution of the underlying data. All data points should be shown for plots with a sample size less than 10. For larger sample sizes, please consider box-and-whisker or violin plots as alternatives. Measures of centrality, dispersion and/or error bars should be plotted and described in the figure legend.*

We did these changes to the existing figures and formatted the new figures and panels accordingly.

Reviewer #1 (Remarks to the Author):

*In their manuscript, Vettkötter et al. describe the development and application of a new optogenetic tool "optoSynC" that enables neuronal inhibition via light induced clustering of synaptic vesicles. The method relies on homooligomerization of AtCRY2, a blue light photoreceptor, targeted to vesicles using fusion to synaptogyrin (SNG-1). The authors proof the functionality by various methods in the nematode *C. elegans*, namely by behavioral readouts, electrophysiology, TEM imaging and further demonstrate applicability in different neuronal cell types as well as in a spectral multiplexing experiment with the red light activated channelrhodopsin Chrimson. The carried out experiments appear well performed and always include appropriate controls. However, as the authors have notified and discussed, a similar approach has been described recently (Won, J. et al. Opto-vTrap, an optogenetic trap for reversible inhibition of vesicular release, synaptic transmission, and behavior. Neuron 110. 2022). Interestingly, the Opto-vTrap resembling construct was not successful in *C. elegans*. The approach presented by Vettkötter et al. is somewhat more simplistic (homooligomerization versus heterooligomerization) and the application is so far limited to *C. elegans* only and would benefit from application in other animal systems or proof of concept experiments in mammalian neurons.*

We thank the reviewer for these recommendations. We now added proof of principle experiments in zebrafish (light-induced suppression of behavior) and murine hippocampal neurons (blockade of synaptic transmission in culture, imaging of SV recycling). These data are shown in the new Figs. 5 and 6.

Further, the authors point out the superiority of optoSynC for presynaptic inhibition, mainly by referring to kinetic properties and reversibility, disproportionately and at multiple passages in their manuscript (detailed below). If comparisons with other tools for presynaptic optogenetic inhibition are desired, they should be performed thoroughly and correctly.

We realize that our writing can be seen as inappropriate. We did not intend to disparage the usefulness and performance of other tools for synaptic inhibition. We now tried to pay more attention to this and point out the different nature of our tool and which applications it may enable that are not addressable with other tools. To compare optoSynC to other silencing tools, we have characterized a range of these tools in previous work in *C. elegans*, so we compared optoSynC parameters to our own previously acquired data in a more systematic way, and we derived parameters of other tools described in the literature (from published data). This comparison, being aware that precise parameters may be different if they would be derived from side-by-side experiments in *C. elegans*, is summarized in a matrix of onset and offset kinetics of the respective tools, now in Figure 9a.

As optoSynC appears to be a versatile and promising tool for presynaptic inhibition, the manuscript should be suitable for publications after revision and addressing the comments detailed below.

1) Title, Abstract, lines 65-68, 96-97, 182, 382-38 and others: *In context of the title as well as the entire manuscript, the authors should be very careful how to discuss and compare temporal terms (fast, rapid, temporal precise, instantaneously, etc.) regarding OptoSynC itself and in comparison with other optogenetic tools.*

See above, we toned down, or rewrote these sections more clearly, and included a comparison of the relevant parameters of other silencing tools.

It is further not clear what the authors are referring to when stating that optoSynC is the fastest tool that does not depend on hyperpolarization. Although GPCR-mediated inhibition can activate K⁺-channels, the mechanism of synaptic inhibition is independent of K⁺-channel activation.

We clarified this point and added the comparison of the time constants (Fig. 9a). Also, we ordered the tools by the mechanism of action they utilize. Here, among tools that do not affect cellular biochemistry or ion concentrations, optoSynC shows the fastest action. The role of GPCRs coupling to G_{i/o} of course is only in part through GIRK channels, and otherwise through effects via protein kinase A. The fast effects of PKA signaling are likely evoked through modulation of synaptic ion channels, though this is not always clear; we adjusted our wording accordingly.

While they judge that GPCR activation has slow onset “dozens of ms to sec”, they describe their tool as “rapid” although the on kinetic is about 15s in the presented behavioral assay.

We revised the discussions about speed and used more precise wording and descriptions. The onset kinetics of optoSynC are probably faster than the 15 s, as this was derived from effects on swimming behavior. We refined this analysis, resolving more time points, showing that it is actually twice as fast. For other behavioral assays (light-evoked escape behavior), blocking of this behavior occurs within the first 2-3 seconds of optoSynC stimulation, with a similar time constant of 6.75 s, as shown in Fig. 2e (ignoring ‘outliers’), compare to 7.23 s for the swimming assay (Fig. 1b, analyzed with more time points than previously).

In the current form of the manuscript the kinetics appear to be heavily oversold at multiple passages, which should be revised in a careful and correct manner.

We revised this.

Further the authors should, also cite at least the recent literature regarding optogenetic GPCR-mediated synaptic inhibition (e.g. Mahn et al. and Coptis et al., both Neuron 2021).

Thank you for pointing this out. We now included these (and other GPCR) citations and included the parameters deduced in these papers in our matrix.

2) optoSynC appears to be a potentially useful optogenetic tool for modulation of the presynapse. However the presented data shows only application in C. elegans. For the broad neuroscience community using optogenetic tools, the manuscript would highly benefit from an application in mammalian neurons. I am aware that this would go beyond the expertise of the Gottschalk lab and therefore listed this as a minor suggestion, but I would still like to emphasize that I would highly recommend it.

We thank the reviewer for this suggestion. We have now performed such experiments, therefore also including new co-authors. optoSynC was expressed in zebrafish neurons, where it could be photoactivated to inhibit light- or touch-evoked escape behavior. Furthermore, we expressed optoSynC in mammalian neurons, addressing evoked synaptic transmission via a pH-sensitive fluorescent protein. Here, pre-illumination of the cultures with blue light abolished the otherwise well observable transmission. These data are now provided in Figs. 5 and 6. They demonstrate the utility of optoSynC in two vertebrate systems, thus, most likely, the optoSynC principle can be applied also in many other systems.

3) Data representation:

a) Much data in this manuscript is presented as bar graphs showing only mean +- SEM. Single point data should be provided wherever possible to allow the reader to estimate the data distribution. Violin

or box plots could be used as an alternative or even better in addition to the current data representation. Especially for $n < 10$ (mPSC data) single data points should be shown.

Thank you, yes, we changed this. For the swimming behavior, we re-analyzed the data, to extract not only mean data, but also data from individual animals (which we now show in violin plots). For crawling, this meant that only tracks that last longer than half the duration of the video are included (the tracker loses identity of the animal if its track crosses with another animal; then it tracks the animal as a new entity), thus avoiding that animals are measured several times. For mean analyses across time intervals, we first averaged the data for these intervals, and then took the mean of the different experiments (generally, $N=3$). Also, we added the single point data for the electrophysiology, and showed them as paired data.

b) X-axis breaks: The authors show initial velocities, followed by axis breaks until the relevant part of their experiments starts/ worms show constant behavior. I think that the initial velocities are not necessary to show here and that the axis could start right from 250/300s of the experiment. The authors can provide the data starting from time point zero at the source data.

Ok, we did this for Fig. 8 (previously Fig. 6), but not for all figures, as the nature of the experiment is that the animals are transferred to the assay plates, which startles them, thus one wants to be sure that they have been accommodated again before the experiment is started. It also serves to show that the animals are not non-specifically affected by expression of the transgene, and, for example in Fig. 2d, are able to crawl fast(er).

c) Y-axis breaks: are not appropriate as they distort the strengths of the observed effect.

This was removed in Fig. 6 (now Fig. 8), but we kept it for the representation of the SV distances in violin plots. (e.g. 4c,f). They help seeing the individual points, and, as the most important difference is in the closest SV distances, it helps visualizing these differences (that would otherwise be obscured in the full scale. The data points that are not visible in Fig. 4c due to the axis break represent a very minor portion ($<6\%$) of the total data points. We provide this for illustration here in the letter:

d) Wherever paired data is shown it preferably should also be presented as such.

We did this for Fig. 3d, f. For other figures, where we follow time courses, it would be impossible to visualize.

4) Figure 3d and f: In the statistics of the mPSC frequency analysis the authors binned data for 30s prior optoSynC activation but only for 10s post blue light activation. The analyzed time frames should be equal. (This should ideally be the case for all statistics / binning done within the manuscript unless there is a qualifying reason not to do so).

We adjusted this and throughout, it did not alter the results.

5) *Statistics & Data:*

a) *In several plots p-values (asterix) as well as the same statistical method are defined multiple times within one figure legend. This should be summarized into one definition per figure where possible.*

Yes, sorry, we now summarized this at the end of each figure legend.

b) *Test statistics (F, t, r) and exact p values are not reported.*

We provide a data table with all the source data for each figure, and included there a summary of these test statistics.

c) *Source data should be reported.*

See above.

6) *Spectral multiplexing experiment in combination with Chrimson: While in its current form the experiment reveals the fast onset of the clustering based inhibition it also demonstrates optical crosstalk between the blue light activated optoSynC and the red light activated Chrimson. The latter is known to be also activated by blue light as also demonstrated by the authors. However, given the high light sensitivity and long-lasting activity of optoSynC the authors should be able to demonstrate crosstalk-free low intensity blue light activation of optoSynC without activating Chrimson.*

We did this experiment, reducing the blue light stimulus intensity further, to a point where Chrimson is no longer activated (now Fig. 8d, e). We also provide the light intensity 'titration' data for optoSynC activation in Supp. Fig. 3, such that the two datasets can be compared.

Ideally the Authors could combine their optoSynC light sensitivity data with measurements of Chrimson activity

See above, data to do this comparison is now available. However, the one dataset is measured with pan-neuronal expression (optoSynC, Supp. Fig. 3) and the other with cell-specific expression in PVD (Chrimson, Fig. 8d), which makes it difficult to directly compare the two datasets. It provides a good guideline, though, which intensities should be used for future multiplexing experiments.

to calibrate for a final experiment demonstrating the above mentioned crosstalk-free synaptic inhibition with upstream Chrimson activation with red light.

See above, the experiment is now presented in Fig. 8e.

7) *The authors should provide a light titration curve and calculate the EC50 of optoSynC from the data that they already have reported in the supplemental material.*

This is now in Supp. Fig. 3b.

Ideally, this should be done in comparison to Chrimson activation with blue light as mentioned under 6).

See above.

The authors could also include a subpanel with the CRY and Chrimson activation/absorption spectra to visualize the source of the potential optical crosstalk.

Yes, we added this to Fig. 8d.

Further, the authors should directly compare the pulsed vs. continuous illumination in the above mentioned sensitivity graph. This information is very valuable for the reader and should not be hiding in the supplement.

We added this information as Supp. Fig. 3d, e.

8) Methods section: “mm” should be replaced by “mM” in the electrophysiology part.

Yes, sorry, this was overlooked. Corrected.

7) The authors have demonstrated reversibility of optoSynC. For some experiments it would be of interest to show the ability to repetitively perform presynaptic vesicle clustering.

We performed an experiment with one repetition, that showed essentially the same activation and recovery characteristics as the first activation episode. This data is shown in Supp. Fig 3a.

Reviewer #2 (Remarks to the Author):

The manuscript by Vettkotter et al. introduces optoSynC, a new optogenetic tool to suppress neuronal activity. The presented strategy is based on photoinduced homo-oligomerization of cryptochrome-2 (CRY2), which in turn is fused to the synaptic vesicle (SV) protein Synaptogyrin (SNG-1). This setting enables light-triggered SV clustering, resulting in reduced neurotransmitter release from the presynaptic active zone and decreased neuronal signalling. As proof-of-principle, the authors carry out electrophysiology, high-pressure freezing electron microscopy, and behavioural analyses in *C. elegans*.

This nicely designed innovative study has been performed elegantly using state-of-the-art technologies. The detailed analyses support the authors’ conclusions and I anticipate that the manuscript will attract considerable attention extending beyond the field of *C. elegans* Neuroscience. My main concern relates to the utility of the newly developed optoSynC approach. A main advantage of optogenetic strategies is the ability to functionally target specific neuronal subpopulations. However, the cell-specific expression of optoSynC appears to provide only approx. 50% activity suppression, as judged by the locomotion assays in Fig. 5a. Similarly, the electrophysiological measurements of mEPSCs at the single cell/muscle level also show only a roughly 50% reduction in frequency. This moderate inhibition, combined with the necessity to express the CRY2-based fusion construct in the *sng-1* mutant background, will likely limit many applications.

This point is well-taken, optoSynC is not perfect in that sense. We want to discuss two points here, however. Compared to tools like the photosensitive degron (psd; Hermann 2015, Curr Biol 25: 749-50) (achievable inhibition 60%) or PA-BoNT (Liu 2019, Neuron 101: 863-75; achievable inhibition 55%), the level of inhibition achievable by optoSynC is not worse (80% in pan-neuronal and 55% in cell-specific expression). It may not to be expected that the animals stop completely, e.g. synaptotagmin mutants were still able to swim with ca. 25% speed compared to wild type (Hermann 2015, Curr Biol 25: 749-50). Yet, when we performed inhibition with ACR2 (targeting membrane potential and not synaptic proteins), this was

indeed complete (now provided in Fig. 9b). It is possible that membrane-potential based inhibition has effects that are further reaching, as the motor neurons are connected by gap junctions to the premotor interneurons (Kawano 2011, *Neuron*, 72, 572–586). This aspect cannot be expected from optoSynC. However, when it is expressed pan-neuronally, it also affects the interneurons, likely leading to the additive effect.

For the electrophysiology experiment, we would like to point out that it is not known how the rate of mPSCs measured in a dissected, immobilized animal compares to the ability of the intact animal to swim. This may not be linear, meaning the observed inhibition by 60% may translate into a larger effect at the level of behavior. Also, the analysis of miniature postsynaptic currents depends on how the definition of a mini is set in the analysis software (we usually use a threshold of 8 pA). If we apply a more stringent criterion (i.e. 10 pA), we achieve an inhibition of ca. 70% (see below), however, it also reduces the basal rate of minis (from 33 Hz to 19 Hz). The former is more in agreement with typical numbers reported in the literature, so we rather stick to this value.

Expression in the *sng-1* mutant background should not be a major limitation, as these mutants are not having any strong phenotype and can be easily crossed. Also, not in all cases that we tested, the *sng-1* mutant background was required, i.e. for cholinergic neurons, the achievable inhibitory effect was not reduced or even a bit stronger in the wild type background, compared to the *sng-1* mutants background (Supp. Fig. 6c).

In the end, we would argue that it depends on the type of question and experiment that researchers are after, to choose one or another tool for inhibition. While ACR2 is faster and more efficient in blocking locomotion activity, we observe that the 30 s activation of ACR2 leads to profound changes in the physiology of the neurons, such that locomotion exhibits a rebound increase, that takes about 1.5 min to recover (see Fig. 9.b). We compare the kinetic characteristics of different tools for inhibition, i.e. ion channels and pumps, metabotropic receptors, clustering tools and tools evoking protein damage at the synapse, and optoSynC is the fastest of the latter two types of inhibitors (Fig. 9a). Unlike the other tools, optoSynC does not alter cellular biochemistry or ion gradients, nor does it damage proteins, thus it is less invasive and recovers much faster. As long as one can clearly detect the effects of inhibition, it may be sufficient to answer specific questions as to the nature of the neural circuit under study. Besides, optoSynC enables to study different SV pools and the temporal aspects of SVs passing from one pool to the next. We also note that, would one be able to target optoSynC to specific subsets of SVs, or to dense core vesicles (DCVs), very interesting possibilities to study synaptic co-transmission would arise.

I fully agree with the authors' explanation of the advantages optoSynC offers over tools such as miniSOG or PA-BoNT. However, for the interested user it would be important to illustrate in more detail why, at least for certain applications, optoSynC could be preferable over light-gated ion channels, which offer tighter temporal control. The authors describe the potential issues associated with variable chloride concentrations and the danger of in fact activating instead of inhibiting cellular processes by photoinduced chloride conductances. I feel the current study would be significantly

strengthened by supporting this argument with a direct comparison between an ACR and optoSynC in an experimental setting.

Compare data from Bergs 2018 and optoSynC paper:

We now added an experiment for ACR2 inhibition of cholinergic neurons in the swimming assay (Fig. 9b, and compare to Fig. 7a; note, the experiments in Bergs et al. 2018 were body contraction assays, which cannot be compared to the optoSynC swimming assays we do here). ACR2 strongly and rapidly blocks swimming, and animals quickly resume swimming after light-off. So obviously ACR2 is more efficient and faster. However, it also has side effects, as the animals show a rebound effect and swim more vigorously than before the stimulus, an effect that takes ca. 90 sec to recover back to the initial levels. Interestingly, this rebound is not immediate, but develops over the first 30 sec, before it levels off. We suggest this may have to do with an altered Cl⁻ gradient in the neurons, due to ACR2 action (it has a very high conductance), and that the cells need to readjust the internal Cl⁻ concentration, which may be a comparably slow process. Currents during this period may contribute to an elevated membrane potential, translating into higher locomotion velocity. Of course, also optoSynC animals do not show normal locomotion after the inhibitory light pulse, but this is due to the slow de-clustering of SVs, which should otherwise not have any effect on cellular biochemistry.

Other than that, I have only minor comments:

- Line 24: "causing approximation". Consider rephrasing for the sake of clarity.

We altered this to "optoSynC clusters SVs, moving them closer together, observable by electron microscopy."

- Figure 1c, d: following photoactivation of optoSnyC does swimming fully recover at later timepoints (>30 min)?

We have extended the duration of this experiment and also added a repeated stimulation / recovery experiment Fig. 1c and Supp. Fig. 3a.

- Line 178: "of blue light", delete "of".

Ok, thank you.

- Figure 4: Upon light-induced SV clustering the mean nearest SV distances drop to approx. 40 nm. Does this distance match the calculated length of the SNG-1-CRY2olig link?

We discussed this in lines 449-453. The distances we measured are edge-to-edge, thus one should not only pay attention to the median distance, but also to the nearest observed distances, which we think may represent the length of the SNG-1::CRY2 link. The CRY2 tetramer, by crystal structure, has a diameter spanning 95-125 Å, which is close to the observed minimal distances of ca. 10 nm. Of course, we don't know how the proteins / aggregates would arrange in 3D around / between the SV membranes, but it is of approximately the right size.

- Figure 6a: consider denoting the light-stimulated conditions (pink and light blue with asterisks) more clearly.

Ok

- Could the authors explain or speculate why the LARIAT approach does not work for them?

We don't know, but one plausible idea is that it is less efficient if two protein species need to meet, and therefore the amount of the soluble protein is critical. Thus, if the untagged/soluble

protein is not delivered in sufficient amounts to the synaptic terminal, it would reduce efficiency. This could be limiting in the tiny *C. elegans* axons (diameter can be < 100 nm) and different in mammalian neurons, with, presumably, larger diameter axons (at least 200 nm, and much larger; Schmidt 2017, Nature 549: 469pp).

- Why do the authors in some cases refer to CIB1 and in other cases to CIBN?

This was confused, we now clarified this. Thank you for pointing it out.

Reviewer #3 (Remarks to the Author):

In this study, the authors report a novel and a much needed optogenetic approach for rapid and reversible silencing of target neurons without affecting neural polarization. The method relies on the light-induced oligomerization of cryptochrome CRY2 which leads to clustering and sequestration of synaptic vesicles. While the silencing is limited in its effectiveness (e.g., some analyses showed less than 50% inhibitory effects), this method still offers a unique and valuable approach that may be employed in many future studies. Furthermore, while the current study demonstrated the applicability of the method in C. elegans neurons only, this method could in principle be adapted for use in other model animals.

We thank the reviewer for these recommendations. Please see above (reviewer #2, p6) for a brief discussion about the comparably low inhibition percentage. We did now test the applicability of optoSynC in two vertebrate species / cells, i.e. zebrafish neurons, and murine hippocampal neurons in culture, with very promising results.

There are several major concerns that the authors should address:

1. Figure 1c: it seems that the optoSynC-expressing animals do not fully recover by the 30 min of experimental data shown in the figure. Do they really eventually fully recover? Could you show the data after the 30 minutes?

Yes, they do, we show this now in an extended Fig. 1c,d and Supp Fig. 3a. Note that non-stimulated animals show a decline of swimming frequency over the time course of 1h, too.

Similarly, in figure 5c: do the GABAergic neurons fully recover? The speed decrease is very low but is it possible to show longer time periods to demonstrate the full recovery?

In Fig. 7c,d, we provide this information. Yes, they do.

2. Figure 2a: It is not clear why the speed of optoSynC animals (wt, cyan and lite-1 bkg, yellow) does not decrease upon light stimulation, just as was noted for the swimming behavior. The speed seems to remain ~constant, arguably merely inhibiting the expected speed increase upon exposure to the blue light. However, I would expect the speed to considerably drop down as shown in figure 1.

This had to do with the high light intensity used to stimulate optoSynC (1 mW/mm²), which also induced a photophobic response. We now repeated this experiment at 10 times lower light intensity (0.1 mW/mm²), where this side effect was absent, and the animals clearly reduced their crawling speed. See Fig. 2d-f.

Could it be that this strain is defective speeding up upon exposure to blue light (despite the fact that optoSynC-DA can speed up)? Perhaps it is possible to use the optoSynC in the lite-1 background and induce a speed up by poking the tail with and w/o light to observe the slow down.

No, given the new experiment, we think this is not a concern. Also, the animals can crawl fast, as one can see in the left end of the graph(s) in Fig. 2a, d. We recorded the animals' speed right from the moment when we placed them on the assay plates (which startles them). The experiment only starts after 5 min, after the animals have accommodated. Also, we tested that the optoSynC animals respond normally to a mechanical stimulus (presented at 300s into the experiment; arrowhead):

3. Figure 3g: To explain the results, the authors suggest that optoSynC inhibits ACh secretion and that is why they are slow to get paralyzed. If this indeed the case, then it is mandatory to show the speed by which the animals moved during these hours (for all strains and conditions). It is postulated that optoSynC animals moved significantly slower than the other animal groups throughout the experimental time frame.

The data shows not actual movement speed, but the ability to move at all. I.e. paralysis is tested by poking the worms and inducing (or not) a movement, for example head twitching. This is counted as 'not paralyzed'. The y-axis shows fraction of tested animals that were moving. In fact, the aldicarb induced paralysis acts much faster at the level of locomotion speed. It is not clear whether this would produce similar results, and our tracking device can only image one plate at a time. We added data for animals in the absence of aldicarb, and with or without blue light, and they can respond normally (100% of the time) to the test, throughout the 6 h of the experiment. This was added to Fig. 3g.

Moreover, continuous blue light for such long durations may affect the animals and their locomotive behavior. It is therefore important to repeat the experiment with light on and in the absence of Aldicarb to assess the effect of light itself.

See above. Done.

4. Figure 4: Analysis of the EM data was used to infer a possible mechanism for the inhibited SV release. However, an important control is missing: The small difference observed in the distances between the SVs could also be attributed to the exposure to blue light that is known to trigger neural activity. It is appreciated that repeating the experiments to compare WT animals in light and dark is a tremendous effort, but could the authors at least provide an explanation to refute such possibility. This is important since if light itself can explain the minute differences, then the entire logic that strives to explain the inhibitory mechanism may collapse.

This is a valid point and we had actually obtained the experiments / HPF-fixed and stained worms for this. We now finished these analyses and are pleased to report that the *sng-1* non-transgenic controls indeed do not show any effect of light on SV distance, or other parameters, apart from the mean distances that docked SVs have along the plasma membrane. This data is now shown in Fig. i-p, Supp. Fig. 4k-r and Supp. Fig. 5i-p.

5. For many of the comparisons in figure 4 and the relevant supplementary figure S3, the difference between – and + light is minute (e.g 2 nm difference on a scale of >30 nm distances). Despite the fact that such differences come out as statistically significant, it looks like this is due to the large sample size (e.g fig 4c and suppl. fig 4a-b). For example, suppl. fig 4c does not come out significant presumably due to the much smaller sample size. Since the mechanism underlying the inhibition relies on such significance differences, its plausibility is questionable.

The fact that TEM data on stained samples is not accurate to the nm requires to look at many individual SVs. In cryo-electron tomography, the more 'particles' one averages, the

more accurate does the electron density map become. Here, similarly, we aim to analyze as many particles (SVs) as possible.

For the SV median distance, the observed difference is 4.6 nm. Given 30 nm SV diameter this is >15%. We also averaged the SV distances per section, and per synapse, to exclude that some systematic difference affects the outcome. Also then, we found highly significant differences. The minimum distance SVs can have is dictated by the CRY2 oligomer size, i.e. they cannot move closer together than this (see above). If one looks at the smallest distances in the distribution, it also becomes quite clear that the SVs move closer together. Last, the SVs do not do this in a non-transgenic control, when dark vs. illuminated samples are compared (see the new panels in Fig. 4, and in Supp. Figs. 3 and 4).

Also, the Mann-Whitney test was used to extract significance. However, this test assumes that the variables between and within the groups are independent. So if EM micrographs were from the same animal, or if different synapses were analyzed from the same neuron, then the extracted variables may not be considered independent.

We do compare numbers between different genotypes or experimental conditions, so these are independent. I am not sure to what extent the variables within one group need to be independent, or can be independent in our case. Strictly, this would mean that only one SV distance per animal could be analyzed – an impossible endeavor given the workload this would require. We considered other tests. An independent t-test – however, this would require the data to be normally distributed, which they are not (still, for a number of datasets we tested, the significances were the same as for the Mann Whitney test; likewise, for a t-test with Welsh's correction). The alternative to a Mann Whitney test is a Wilcoxon Signed Rank Test. However, this test only provides information whether the median of one group of data differs from an expected median, and does not allow to compare two groups. To our knowledge, the Mann Whitney test is commonly used in EM analyses of this kind.

The methods section describes that 20-40 animals were used for each group, but could it be that most of the EM data were eventually collected from a few worms?

This may have been a misunderstanding. We picked 20-40 animals to the freezing planchette, for each condition. From those, two animals, undamaged by the freezing and fixation procedures, were chosen from each planchette for sectioning. Of these sectioned animals, 9-30 different synapses were chosen and the number of sections analyzed, as given in the manuscript, is between 40 and 90.

6. Figure 6a: again, I do not understand why there is no reduction in the basal speed of the animals following exposure to light.

Please see above, we measured this again and used less light, which avoids evoking the photophobic behavior that may have counter-acted slowing induced by optoSynC activation.

7. Since optoSynC is sensitive to very low intensities of blue light, is it also sensitive to ambient laboratory-typical light? Were such controls conducted? For example, ideally, should all handling with these transgenic animals be done in the dark?

Yes, ambient light may be able to evoke pre-activation of optoSynC (see new light-titration curve in Fig. 3e), though only at a low level. Nevertheless, all animals were kept in the dark until the actual experiment, and in these, we used red filtered light as background illumination, to avoid unwanted activation of optoSynC.

8. Overall, this new tool seems to provide only partial inhibition. For example, ~50% reduction in mPSC frequency.

50% and 20% reduction in the swimming cycles when expressed in cholinergic and GABAergic neurons, respectively.

But it is almost 90 % when the tool is expressed pan-neuronally. Please see our discussion also further above. It is possible that the nervous system we analyze, i.e. driving locomotion, which is not only affected by ACh and GABA, but also by electrical coupling, may be preventing 100% inhibition, as we achieve with ACR2. In experiments in which one does not want to temper with ionic concentrations and affect membrane currents to evoke inhibition, optoSynC is a very useful alternative.

With this caveat, the tool may not be suitable to various applications. Will be good if the authors acknowledge that and discuss which experimental designs may benefit using this tool at its current fair efficiency.

We altered our discussion and extended this with Fig. 9a. We hope we improved this point and made it clear where we see the strength and weaknesses of the optoSynC approach.

Minor issues to be addressed:

1. The data presented in Supple fig 1 is presumably using a pan neuronal expression but it is not indicated in the text nor in the legend.

Yes, this was adjusted.

2. Suppl. fig. 2c. I believe the x-axis units are minutes (not sec).

Yes, thank you for spotting this.

3. In line 269, values in the parentheses are provided as a range but it is not specified what is this range. This is just an example as such value ranges repeat throughout the text.

We now explained this better, and at various places in the text / legends.

4. the one-phase decay fit does not seem to be accurate. It is evident from supplementary figure S2d that while the two decaying curves are very similar, the resulting tau values are very different. This is because the decay function actually proceeds past the relevant time period shown in the graph. To extract more accurate tau values, perhaps the authors should consider to include additional data points that are beyond the specified time window. The same holds for the tau computed in figure 1b.

This was reanalyzed, it is now shown in Supp. Fig. 3b. Also Fig. 1b was analyzed anew, after increasing the sampling to shorter time intervals. Thank you for this suggestion.

5. Supplementary figure S2b: is there an explanation why animals expressing the GFP-tagged version of the protein show increased swimming cycles after being exposed to blue light?

Does the reviewer refer to the small increase during / before the last stimulus?

We are not sure, maybe this is due to overall inefficient clustering, or some startling response that is able to overcome the inhibition. But this is pure speculation.

6. In supplementary figure 4, maybe instead of showing the absolute values of DVs, it would be better to show the ratio between docked and undocked vesicles?

Does the reviewer refer to S3g (now S5e)? We kept to what is typically done in these types of analyses. Since the size of synapses / synapse cross sections can differ, we relate the number of docked SVs to the area which is available for docking, and not all sections contain DVs. Of course one may argue that the amount of SVs in the bulk phase may also influence how many of those can be mobilized to dock. But the reserve pool of SVs would have to be included in its three dimensions, to estimate the amount of SVs (which we cannot always do). Since the *en-passant* synapses are mostly cylindrical, the length of the membrane can be more easily estimated from a single section.

We did this ratio analysis, and it made no difference for the optoSynC animals, however, for the *sng-1* controls it did:

We can provide this analysis, if this is wanted, as additional panels in Sup. Fig. 5 - but it is not clear to us how to interpret this result. Therefore, we suggest to keep the figure as it is.

REVIEWERS' COMMENTS

Reviewer #1 (Remarks to the Author):

In the revised version of their manuscript, the authors have faithfully addressed the majority of my concerns raised in the previous review round. In particular, they now show successful inhibition in zebrafish and murine neurons beyond the application in *C. elegans*. Therefore, I now recommend the manuscript for publication.

Sincerely, Jonas Wietek

Reviewer #2 (Remarks to the Author):

My comments and concerns have been adequately addressed by the authors. I support publication of the manuscript in its present form.

Reviewer #3 (Remarks to the Author):

The authors have greatly improved the manuscript and addressed all my initial concerns.